# Lower boundary conditions in Land Surface Models. Effects on the permafrost and the carbon pools: a case study with CLM 4.5.

Ignacio Hermoso de Mendoza[1], Hugo Beltrami[2], Andrew H. MacDougall[2], and Jean-Claude Mareschal[1]

[1]Centre de Recherche sur la dynamique du système Terre (GEOTOP), Université du Québec à Montréal (UQAM), Montréal, Québec, Canada
[2]Climate & Atmospheric Sciences Institute and Department of Earth Sciences, St. Francis Xavier University, Antigonish, Nova Scotia, Canada

*Correspondence to:* Hugo Beltrami (hugo@stfx.ca)

**Abstract.**

Earth System Models (ESMs) use bottom boundaries for their Land Surface Model (LSM) components which are shallower than the depth reached by surface temperature changes in the centennial time scale associated with recent climate change. Shallow bottom boundaries reflect energy to the surface, which along with the lack of geothermal heat flux in current land

surface models, alter the surface energy balance and therefore affect some feedback processes between the ground surface and the atmosphere, such as permafrost and soil carbon stability. To evaluate these impacts, we modified the subsurface model in the Community Land Model version 4.5 (CLM4.5) by setting a non-zero crustal heat flux bottom boundary condition uniformly across the model and by increasing the depth of the lower boundary from 42.1 m to 342.1 m. The modified and original land models were run during the period 1901-2005 under the historical forcing and between 2005-2300 under forcings for two

future scenarios of moderate (RCP 4.5) and high (RCP 8.5) emissions. Increasing the thickness of the subsurface by 300 m increases the heat stored in the subsurface by 72 ZJ ($1\,\mathrm{ZJ} = 10^{21}$ J) by year 2300 for the RCP 4.5 scenario and 201 ZJ for the RCP 8.5 scenario (respective increases of 260% and 217% relative to the shallow model), reduces the loss of near-surface permafrost area in the Northern Hemisphere between 1901 and 2300 by 1.6%-1.9%, reduces the loss of intermediate-depth permafrost area (above 42.1 m depth) by a factor of 3-5.5, and reduces the loss of soil carbon by 1.6%-3.6%. Each increase

of $20\,\mathrm{mW}\,\mathrm{m}^{-2}$ of the crustal heat flux increases the temperature at 3.8 m (the soil-bedrock interface) by $0.04 \pm 0.01$ K. This decreases near-surface permafrost area slightly (0.3-0.8%) and produces local differences in initial stable size of the soil carbon pool across the permafrost region, which reduces the loss of soil carbon across the region by as much as 1.1%-5.6% for the two scenarios. Reducing subsurface thickness from 42.1 m to 3.8 m, used by many LSMs, produces a larger effect than increasing it to 342.1 m, because 3.8 m is not enough to damp the annual signal and the subsurface closely follows the air temperature.

We determine the optimal subsurface thickness to be 100 m for a 100 yr simulation and 200 m for a simulation of 400 yr. We recommend short term simulations to use a subsurface of at least 40 m, to avoid the perturbation of seasonal temperature propagation.

# 1 Introduction

In the current context of anthropogenic climate change, there is a need to forecast future impacts of climate change as reliably as possible. Climate change projections are based on simulations from ensembles of Earth System Models (ESMs), numerical models of oceans, atmosphere, land, ice, and biosphere subsystems coupled together (Stocker et al., 2013). Modeling of the land system has mainly focused on the interactions between the land surface and the atmosphere (Pitman, 2003), including biogeochemical cycles taking place in the shallow subsurface or soil, such as carbon dynamics (Ramanathan and Carmichael, 2008), soil moisture (Seneviratne et al., 2010), vegetation cover and land use (Bonan, 2008), and surface processes such as albedo and snow cover (Hansen and Nazarenko, 2004). In these Land Surface Models (LSMs) the bedrock layer present below soil is impermeable, and when explicitly modeled, the only process taking place in bedrock is thermal diffusion.

Thermal diffusion in the subsurface allows the land system to act like a heat reservoir, contributing to the thermal inertia of Earth's climate. However, this contribution is relatively small as the capacity of the oceans to absorb energy is orders of magnitude above that of the continents (Stocker et al., 2013). Estimates of the energy accumulation during the second half of the 20th century in the land system show that the heat stored in continents ($9 \pm 1$ ZJ, where $1\,\mathrm{ZJ} = 10^{21}$ J) is less than the uncertainty on the heat stored in oceans during the same period ($240 \pm 19$ ZJ) (Beltrami et al., 2002; Levitus et al., 2012; Rhein et al., 2013). This allows many ESMs to only consider the land subsurface to the shallow depth ($3-4$ m) needed for soil modeling (Schmidt et al., 2014; Wu et al., 2014) and to neglect the bedrock entirely. Still, the thermal regime of the subsurface affects the energy balance at the surface, which in turn influences the surface and soil processes with a feedback on the climate system. Energy variations at the land surface propagate underground, and the use of a too shallow subsurface in land models implies that these signals are reflected towards the surface, altering its energy balance (Smerdon and Stieglitz, 2006; Stevens et al., 2007; Melo-Aguilar et al., 2018; Steinert et al., 2018).

Several works (MacDougall et al., 2008, 2010) have pointed out that, for the long time scales of climate change, the temperature variations at the land surface propagate much deeper than the depths considered in current LSMs, which range between $\sim 3.5$ m (Schmidt et al., 2014; Wu et al., 2014) and $42$ m (Oleson et al., 2013). Theoretical estimates (MacDougall et al., 2008) of heat stored by the subsurface show a difference of one order of magnitude between models using subsurface thicknesses of $10$ m and $600$ m. This suggests that the reflected energy in shallow land models affects the surface energy balance in the simulations, and current ESMs should use land models sufficiently deep for the length of the simulations, to avoid bottom boundary effects on the thermal profiles.

Most of the current land models use a zero heat flux as thermal boundary condition at their base, as the geothermal gradient is small ($\sim 0.02$ K/m) and does not affect temperature much at shallow depth (Jaupart and Mareschal, 2010). Subsurface models that increase the depth of the bottom boundary to hundreds of meters must include the geothermal gradient to properly represent the thermal regime of the subsurface. This can be easily done by using a fixed crustal heat flux as bottom boundary condition of the LSM, as a few models already do (Avis et al., 2011).

Soils in permafrost regions act as a long-term carbon sink that stores an estimated 1100-1500 GtC of organic carbon, twice the carbon content of the pre-industrial atmosphere (MacDougall and Beltrami, 2017; Hugelius et al., 2014). The feedback

between climate and permafrost thawing and associated carbon emissions is expected to accelerate global warming (Schuur et al., 2015). Rising temperatures at high latitudes induce the thawing of permafrost, leading to decay of frozen organic matter and the release of $CO_2$ and $CH_4$ into the atmosphere. Because of the potential positive feedback between thawing permafrost and the climate system, ESMs endeavor to make robust forecasts of permafrost extent and retreat.

The generation of ESMs used in the fifth phase of the Climate Model Intercomparison Project (CMIP5) show large disagreements in the simulation of present-day permafrost extent. The response of permafrost area to the increase of global temperatures shows a wide range of sensitivities across the different CMIP5's LSMs ($0.75 - 2.32 \times 10^6$ km$^2$/K), which in terms of relative losses of permafrost area range between 6% to 29% per K of high-latitude warming (Slater and Lawrence, 2013; Koven et al., 2013b). These differences arise partly from biases in air temperature and snow depth in some models, but mostly from struc-
tural weaknesses of the land models that limit their skill to simulate subsurface processes in cold regions (Koven et al., 2013b; Slater and Lawrence, 2013). Most of these land models rely on very shallow ($\sim$3-42 m) subsurface modules (Cuesta-Valero et al., 2016). We expect that, both the thickness of the subsurface and setting a value of heat flux representative of average continental regions as bottom boundary condition, will affect the evolution of permafrost in a warming scenario, and therefore the release of permafrost carbon.

It is possible to use analytical methods to estimate the effect that the bottom boundary depth and basal heat flux condition have on the thermal profile of the ground (Stevens et al., 2007). Because of the complexity of the biogeochemical processes in the soil, only numerical simulations can estimate how permafrost dynamics and permafrost carbon content are affected by the changes in the thermal profiles. Previous studies with the Community Land Model version 3 (CLM3) have pointed out that, to obtain a realistic representation of permafrost, the model's soil needs to be deep enough ($\sim$ 30 m) to at least reach the depth
needed for the damping of the annual surface temperature signal. Alexeev et al. (2007) used a slab of varying thicknesses (30, 100 and 300 m) at the bottom of a several layers representing the soil at a high resolution, in order to allow sufficient depth to absorb decadal to centennial signals. Nicolsky et al. (2007) used additional soil layers to increase the thickness of the model to 80 m, which they applied at specific locations of deep permafrost. Lawrence et al. (2008) tested soil depths up to 125 m by adding extra bedrock layers, and determined how this affected the extent of near-surface permafrost. However, these studies
did not consider the crustal heat flux, and did not study further effects on the carbon pool. In this paper, we study the effect of increasing the lower boundary depth and adding a geothermal heat flux at the base of the Community Land Model version 4.5 (CLM4.5) (Oleson et al., 2013), which is the deepest (42.1 m) LSM used in the CMIP5 (Stocker et al., 2013). In this paper, our aim is to investigate and quantify the effects of two unrealistic assumptions made by most land models: the shallow depth of the lower boundary and the lack of crustal heat flux in it. We do so by increasing the lower boundary depth and by adding
a uniform geothermal heat flux at the base of the CLM4.5 (Oleson et al., 2013). We investigate the effect of these changes on the permafrost and the carbon pools of the Northern Hemisphere. We also reduce the thickness of the subsurface in CLM4.5 to 3.8 m, to study its effects on the soil carbon pool. To investigate these effects, we carried out simulations between 1901 CE and 2300 CE, using historical climate reconstruction between 1901 and 2005 (Viovy, 2018) and compared two alternative scenarios of moderate and high radiative forcings between 2006 and 2300 (Thomson et al., 2011; Riahi et al., 2011).

## 2 Theoretical analysis

The Earth's continental lithosphere ($> 100$ km) can be considered as a semi-infinite solid for the centennial and millennial time scales considered in the future projection of climate. For a purely-conductive thermal regime of the subsurface, the propagation of a temperature signal at the surface into the ground is governed by the heat diffusion equation in one dimension (Carslaw and Jaeger, 1959):

$$\frac{\partial T}{\partial t} = \kappa \frac{\partial^2 T}{\partial z^2} , \tag{1}$$

where $\kappa$ is thermal diffusivity. The solution of Eq. (1) for a step change $T_0$ in surface temperature at $t = 0$ yields the temperature anomaly at depth $z$ and at time $t$:

$$T(z,t) = T_0 \operatorname{erfc}\left(\frac{z}{2\sqrt{\kappa t}}\right) . \tag{2}$$

The general solution for any surface temperature perturbation $T_0(t)$ starting at $t = 0$ can be obtained as the convolution in time of $T_0(t)$ and the Green function associated to Eq. (1) and the boundary conditions. As the Green function is the solution to a Dirac's delta, it is obtained as the general solution in the time derivative of the solution to the step function in Eq. (2). Therefore, the general solution is:

$$T(z,t) = \frac{z}{2\sqrt{\pi\kappa}} \int_0^t T_0(\xi)(t-\xi)^{-3/2} \exp\left(-\frac{z^2}{4\kappa(t-\xi)}\right) d\xi . \tag{3}$$

Future scenarios (Van Vuuren et al., 2011) predict rising atmospheric temperatures during the present century (Cubasch et al., 2013) with a wide margin of variability and uncertainty. We can represent this future rise in temperatures by a linearly increasing surface temperature $T_0(t) = mt$, with $m$ being the rate of temperature increase. For such surface temperature function, the solution to Eq. (1) is:

$$T(z,t) = mt \left[ \left( 1 + \frac{z^2}{2\kappa t} \right) \operatorname{erfc}\left(\frac{z}{2\sqrt{\kappa t}}\right) - \frac{z}{\sqrt{\pi\kappa t}} \exp\left(\frac{-z^2}{4\kappa t}\right) \right] . \tag{4}$$

Numerical models, however, cannot simulate the subsurface as a semi-infinite solid, also known as half space model, but instead limit the subsurface to a given depth, that varies between models. Many land models include only the upper $3 - 4$ m of the subsurface, which they consider as soil, to model the most basic hydrological processes such as infiltration and runoff in a first-order approximation. Other models further extend the subsurface to include the bedrock below, the deepest currently being the CLM4.5 with a total depth of 42.1 m. We can simplify these models by considering conduction only and modeling the land subsurface as a solid bounded by two parallel planes (the surface and the lower boundary). Assuming a lower boundary condition of no heat flux (as most current models do) and a linearly increasing temperature increasing linearly with time

$T_0(t) = mt$ as surface boundary condition, we obtain the following solution to Eq. (1) (Carslaw and Jaeger, 1959):

$$T(z,t) = m \sum_{n=0}^{\infty} (-1)^n \left\{ \left( t + \frac{(2nd+z)^2}{2\kappa} \right) \mathrm{erfc}\left( \frac{2nd+z}{2\sqrt{\kappa t}} \right) - (2nd+z)\left( \frac{t}{\pi\kappa} \right)^2 \exp\left( -\frac{(2nd+z)^2}{4\kappa t} \right) + \right.$$
$$\left. + \left( t + \frac{(2(n+1)d-z)^2}{2\kappa} \right) \mathrm{erfc}\left( \frac{(2(n+1)d-z)}{2\sqrt{\kappa t}} \right) - (2(n+1)d-z)\left( \frac{t}{\pi\kappa} \right)^2 \exp\left( -\frac{(2(n+1)d-z)^2}{4\kappa t} \right) \right\}, \quad (5)$$

where $d$ is the depth of the bottom boundary. Neglecting near-surface processes such as hydrology or snow isolation, the temperature of the subsurface is described by Eq. (5).

Using Eqs. (4) and (5), we can estimate the effect of the thickness of the model. We have calculated the profiles of temperature perturbation for a rate of surface temperature increase of $0.01$ K yr$^{-1}$, assuming a thermal diffusivity of $\kappa = 1.5 \times 10^{-6}$ m$^2$ s$^{-1}$ (used for bedrock in the CLM4.5 (Oleson et al., 2013)). This temperature increase is within the range of global temperature projections for the 21st century (Collins et al., 2013).

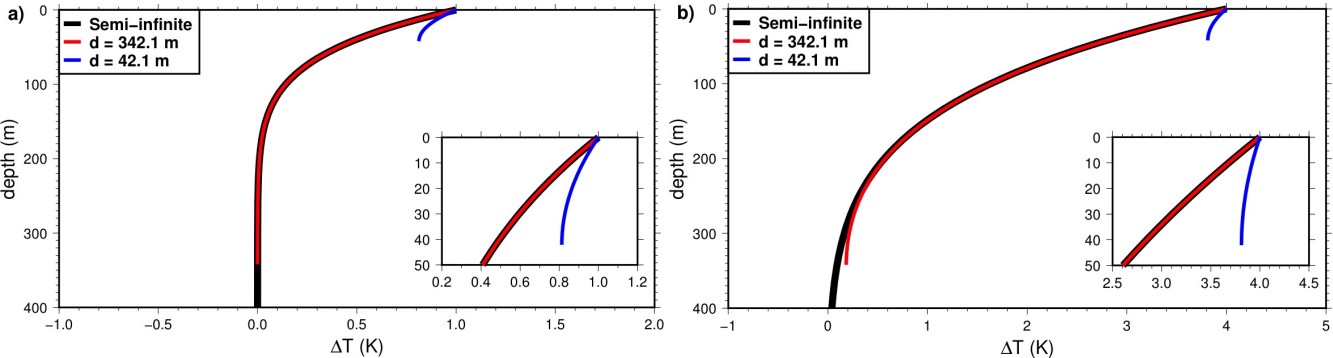

**Figure 1.** Departure from the initial temperature profile due to constant rate of surface temperature increase of $0.01$ K yr$^{-1}$. Analytical solutions for the half space model (black), and for the finite thickness model with bottom boundary at 42.1 m (blue) and at 342.1 m (red). a) Temperature anomaly after 100 yr. b) Temperature anomaly after 400 yr.

We calculated the temperature anomalies for the half space model and the layers of thickness 42.1 m and 342.1 m, after 100 yr and 400 yr. After 100 yr the temperature anomaly for the thinnest (42.1m) model has departed from that of the half space model (Fig. 1a), while the thickest (342.1 m) model cannot be distinguished from the half space solution after 100 yr. After 400 yr the thickest model only has small departure near the base (Fig. 1b). Thus, the response of a model of finite thickness approaches that of the half space model, as long as the bottom boundary is deep enough for the difference between Eqs. (4) and (5) to be negligible.

The maximum time before the shallow bottom boundary affects the thermal behavior of the model is better appreciated in terms of heat absorption by the subsurface. The heat stored in the subsurface can be calculated from the temperature change in Eq. (5) by assuming a uniform volumetric heat capacity $c = 2 \times 10^6$ J m$^{-3}$ K$^{-1}$ (value used for bedrock in the CLM4.5).

The heat absorbed per unit of area for the 42.1 m model is slightly smaller than that of the half space model after 100 yr and less than half after 400 yr, while for the 342.1 m model no difference can be observed (Fig. 2a). The heat absorbed after 100

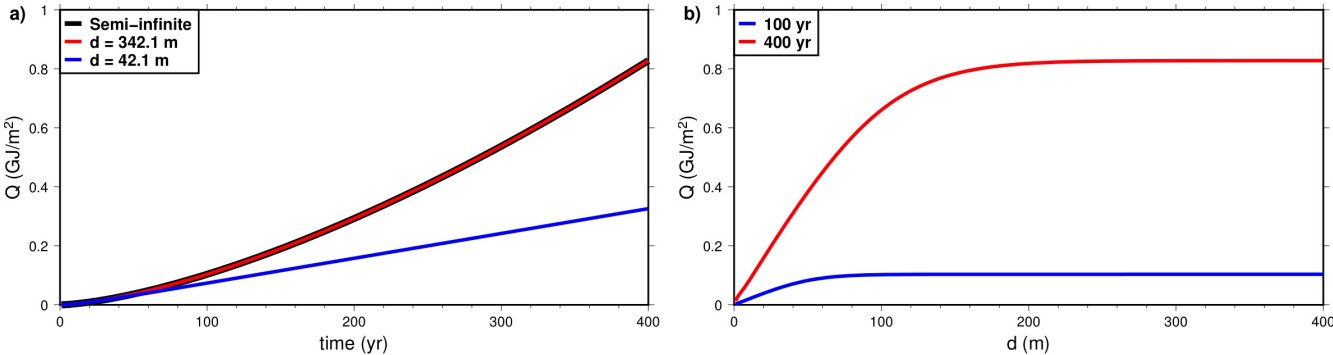

**Figure 2.** Heat absorbed by the land column per unit of area (Q), following the start of a linear surface temperature increase of $0.01$ K yr$^{-1}$. a): Q as a function of time for the half space model and two models of finite thicknesses 42.1 m and 342.1 m. b): Q as a function of the thickness $d$ of the finite model, at 100 yr and 400 yr.

yr or 400 yr increases with the thickness of the model, but reaches a plateau where further increase in thickness does not affect heat storage (Fig. 2b). A bottom boundary depth of 342.1 m is enough for a simulation lasting 400 yr. A bottom boundary depth $d = 100$ m is enough for a simulation of 100 yr, as the heat absorbed by the land column does not rise much with further increasing $d$. A simulation of 400 yr, 4 times longer, needs a bottom boundary depth of $d = 200$ m, only twice as much as 100

yr (Fig. 2b).

The heat equation (1) shows a scaling relationship between distance $d$ and time $t$, $d \propto \sqrt{\kappa t}$. This relation can be used as a first order estimate of the depth where the lower boundary does not affect the thermal profiles for a given duration of the simulation and a value of the thermal diffusivity $\kappa$.

## 2.1   Geothermal gradient

In the conductive regime described by Eq. (1), the subsurface temperature at a depth $z$ is the superposition of the geothermal temperature gradient and the temperature perturbation $T_t$ induced by a time-varying temperature signal at the surface:

$$T(z,t) = T_0 + q_0 \frac{z}{\lambda} + T_t(z,t), \tag{6}$$

where $T_0$ is the the mean surface temperature, $q_0$ is the geothermal heat flux. $z/\lambda$ is the thermal depth and $\lambda$ is the thermal conductivity of the subsurface.

The propagation into the subsurface of an harmonic temperature signal such as the annual air temperature cycle is characterized by exponential amplitude attenuation $\exp(-\sqrt{\frac{\omega}{2\kappa}}z)$ (Carslaw and Jaeger, 1959), where $\omega$ is the frequency of the signal and $\kappa$ is the thermal diffusivity. At depths of $3-4$ m, the amplitude of the annual signal is several degrees. Given the small values ($\approx 0.02$ K m$^{-1}$) of the geothermal temperature gradient in the continents (Jaupart and Mareschal, 2010), the temperature near the surface is dominated by the surface signal $T_t$. Therefore it may seem reasonable to neglect the geothermal gradient

for a thin subsurface layer used in land models (Schmidt et al., 2014; Wu et al., 2014). However, the geothermal temperature gradient can still be influential, even at shallow depths, for temperature-sensitive regimes of subsurface such as permafrost,

and it is necessary to determine the lower limit of permafrost. In the case of the CLM4.5 with a subsurface thickness of 42.1 m, the temperature at the bottom of the model is increased by $\sim 0.84$ K by a geothermal gradient of $0.02$ K m$^{-1}$. If we were to further increase the thickness of the subsurface, the temperature at the bottom of the model would rise proportionally.

## 3  Methodology

### 3.1  Original Land Model

The Community Earth System Model version 1.2 (CESM1.2) is a coupled ESM, consisting of components representing the atmosphere, land, ocean, sea-ice and land-ice. Individual components can be run separately, taking the necessary inputs from prescribed datasets. Because running the coupled model is computationally expensive, we have run only the LSM CLM4.5 (Oleson et al., 2013), forced with prescribed atmospheric inputs (Viovy (2018); Thomson et al. (2011); Riahi et al. (2011), see section 3.5.2). These inputs are precipitation, solar radiation, wind speed, surface pressure, surface specific humidity, Surface Air Temperature (SAT) and atmospheric concentrations of aerosols and $CO_2$.

Carbon and nitrogen cycles are included in the CLM4.5 through the BioGeoChemistry (BGC) module, which includes a methane module (Riley et al., 2011). CLM4.5-BGC can be run at several spatial resolutions. We have used the intermediate resolution $1.89°\text{lat} \times 2.5°\text{lon}$ as the trade-off between resolution and computational efficiency. We used the default timestep of 30 minutes (Kluzek, 2013).

The subsurface is discretized in 15 horizontal layers with exponentially increasing node depths:

$$z_i = f_S \left\{ \exp[0.5(i - 0.5)] - 1 \right\}, \tag{7}$$

where $f_S = 0.025$ m is the scaling factor. Layer thickness $\Delta z_i$ is:

$$\Delta z_i = \begin{cases} 0.5(z_1 + z_2) & i = 1 \\ 0.5(z_{i+1} - z_{i-1}) & i = 2...14 \\ z_{15} - z_{14} & i = 15 \end{cases} \tag{8}$$

The total thickness of the model is 42.1 m. The upper 10 layers, to a depth of 3.8 m, are soil layers where biogeochemistry and hydraulic processes take place. The lower 5 layers are the bedrock, where the only process is thermal diffusion. The soil in each land column has a vertically-uniform clay/sand/silt composition and a vertically-variable carbon density which determines its hydraulic properties and, along with its time-varying water content, its thermal properties. Bedrock layers, assumed to be made of saturated granite (without pores or interstices that could absorb water), are uniform both horizontally and vertically. The thermal properties for bedrock in CLM4.5 are a thermal conductivity $\lambda = 3$ W m$^{-1}$ K$^{-1}$ and a volumetric heat capacity $c = 2 \times 10^6$ J m$^{-3}$ K$^{-1}$, which give a thermal diffusivity $\kappa = \lambda/c = 1.5 \times 10^{-6}$ m$^2$ s$^{-1}$ (Oleson et al., 2013).

As the horizontal dimensions of the grid are much larger than the thickness of the subsurface, horizontal heat conduction is considered negligible and thermal diffusion is considered only in the vertical direction as described in Eq. (1). The land

subsurface is thermally forced at the surface by its interaction with the atmosphere through latent and sensible heat fluxes, and short and longwave radiation. At the bottom boundary, the model uses a zero heat flux condition, which we will modify to experiment with several values of geothermal heat flow.

The hydrology model in CLM4.5 parameterizes interception, throughfall, canopy drip, snow accumulation and melt, water transfer between snow layers, infiltration, evaporation, surface runoff, subsurface drainage, redistribution within the soil column, and groundwater discharge and recharge. The vertical movement of water in the soil is determined by hydrological properties of the soil layers, which can be altered by their ice content as this reduces the effective porosity of the soil. The model also includes an artificial aquifer with a capacity of 5 m below the soil column, from which discharge is calculated. This aquifer is treated as a virtual layer, because it does not interact with the bedrock and it does not simulate any physical process, except for acting as a storage of water percolated from the soil, and draining water to the river transport model.

The parametrization of snow in CLM4.5 follows the approaches of Anderson (1976), Jordan (1991) and Yongjiu and Qingcun (1997). The snow consists of up to 5 layers, whose number and thickness increase with the thickness of the snowpile. Thermal conduction in these layers works like in soil layers, with the thermal properties of ice and water. The model includes fractional snow cover following the method of Swenson et al. (2012), and phase transitions between ice and water in the soil and snow layers.

## 3.2 Carbon model

The Community Land Model version 4 (CLM4) includes a representation of the carbon and nitrogen cycles (CLM4CN) largely based on the ecosystem process model Biome-BGC (Biome BioGeochemical Cycles) (Koven et al., 2013a; Running and Hunt, 1993), which is an extension of the previous model Forest-BGC (Running and Gower, 1991). Forest-BGC simulates water, carbon, and nitrogen fluxes in forest ecosystems, which Biome-BGC expanded with more mechanistic descriptions of photosynthesis and by including more vegetation types in its parameterizations. Later versions of Biome-BGC (Thornton et al., 2002) developed the mechanistic calculations of carbon and nitrogen cycles in the soil, control of photosynthesis by nitrogen, differentiation of sunlit/shaded canopies, calculation of fire and harvest, and regrowth dynamics.

In CLM4.5 (Oleson et al., 2013), we work with the BGC carbon model (Riley et al., 2011). The BGC model expands the Carbon-Nitrogen (CN) model by adding a module of production, oxidation and emission of methane. CLM4.5 also includes updates to photosynthesis, vegetation and hydrology in CLM4. This improves carbon treatment in CLM4.5-BGC significantly over CLM4CN.

As the flow-chart in Fig. 3 shows, there are three main carbon pools in CLM4.5-BGC: the vegetation, the litter (and coarse wood debris), and the soil organic matter (or soil carbon). These pools are subdivided into several sub-pools. The vegetation has distinct pools to account for the different tissues of the plants: leafs, dead/live stems, live/dead coarse roots, fine roots, and a internal storage pool (from where plants can take carbon when they can not photosynthesize). Litter and carbon are each defined in the same 10 horizontal soil layers used for hydrology, and with 3 separate pools each (corresponding to increasingly recalcitrant forms of carbon) arranged as a converging cascade from coarse wood to litter to soil, a structure known as the Century Soil Carbon pool structure (Oleson et al., 2013).

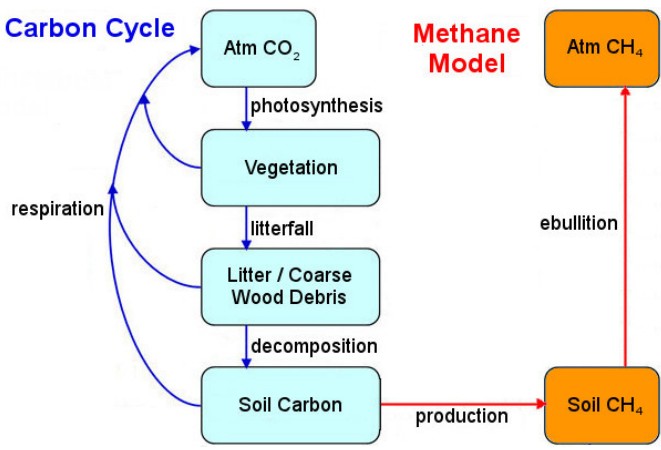

**Figure 3.** Schema of the carbon flux in CLM4.5-BGC. Figure redrawn from Oleson et al. (2013).

The methane model (Fig. 3) produces $CH_4$ in the anaerobic fraction of the soil in a land cell (which can be fractionally inundated in CLM4.5), that consists of the entire soil in the inundated portion of the land cell, and the fraction of soil bellow the water table in the non-inundated portion. The $CH_4$ is produced in the inundated soil where it stays for a short time until it evaporates into the atmosphere (Wania et al., 2010). Thus, the production of methane is closely correlated with the hydrology model. In the CLM4.5 hydrology model, the land can store water within the soil (with a thickness of 3.8 m globally, but variable hydrological properties depending on composition) and in an unconfined aquifer with a capacity of 5 m globally, treated as a virtual layer (which does not interact with the subsurface other than to store water) beneath the soil. In reality, soil thickness is highly variable worldwide, reaching depths of hundreds of meters in some areas, while the global mean is estimated to be $\approx 13$ m (Shangguan et al., 2017).

### 3.3 Modifications of the original model

We made two main modifications to the LSM. First, we increased the thickness of the bedrock and the depth of the lower boundary. Second, we assumed uniform and constant heat flux as bottom boundary condition. Increasing the thickness of the LSM is necessary to reduce the effect of the lower boundary on the temperature profile. The non-zero heat flux adds the geothermal gradient to the temperature profiles of the subsurface, which is needed to determine the lower limit of permafrost in the land column.

We increased the thickness $d$ of the subsurface by progressively adding new layers of constant thickness at the bottom of the land column, to obtain a set of model versions with increasing values of $d$. The thickness of the added layers must be small to fine tune the depth of the bottom boundary. However, the size of the set is limited by our computational resources, as we aim to increase the depth of the bottom boundary by several hundred meters. As a compromise, we used 12.5 m as the thickness of these new layers. The value of $d$ in the original model is 42.1 m (no additional layers) and its highest value is 342.1 m (24 additional layers, with a total thickness of 300 m). In addition, we created a model of reduced thickness $d = 3.8$ m

by eliminating all the bedrock layers in CLM4.5. This does not affect hydrology or any process other than thermal diffusion, because the aquifer is a virtual layer and it does not interact with the bedrock layers.

The bottom boundary condition of the LSM is changed to a worldwide uniform value of heat flux. While the continental heat flux is spatially variable, we lack heat flux measurements in wide areas of the world such as South America, Asia and Africa and the Northern Hemisphere permafrost regions. We use several values of heat flux 0, 20, 40, 60 and 80 $\mathrm{mW\ m^{-2}}$ to cover the range of heat flow values observed in stable continents (Jaupart and Mareschal, 2010).

## 3.4 Permafrost treatment

We define a subsurface layer as permafrost if it remains 2 consecutive years below 0 °C. This definition does not account for the water/ice content of a layer, as we also want to define permafrost in the bedrock layers where no water is present. As the ice content in the soil hinders the movement of liquid water within it, permafrost is closely linked with the hydrology model.

Near surface permafrost is commonly defined as the permafrost present within the upper 3 m of the soil (Nicolsky et al., 2007; Koven et al., 2011; Schuur et al., 2015), but this depth can be different for some land models where the soil depth is larger than 3 m (Lawrence and Slater, 2005). As in CLM4.5 the soil layers make the upper 3.8 m of the land column, we define near-surface permafrost as the permafrost present above this depth.

Because natural soils can reach deeper than the 3.8 m used in CLM4.5, we aim at gaining some insight on how bottom heat flux and model thickness affect permafrost deeper than 3.8 m. However, it is outside the scope of this study to introduce a realistic soil thickness in CLM4.5. For this reason we will also study the permafrost present between the surface and a depth of 42.1 m, the thickness of the CLM4.5 subsurface, which we define as intermediate-depth permafrost.

While near-surface permafrost and intermediate-depth permafrost define permafrost within a depth range, to study the maximum depth of the top of the permafrost we use the concept of Active Layer Thickness (ALT). In environments containing permafrost, the active layer is the upper layer of soil that thaws during summer. The ALT is the maximum depth at which annual temperature variations at the surface are able to thaw the soil, which coincides with the upper limit of permafrost. ALT provides information on permafrost complementary to its areal extent, as variations in the thermal regime of the subsurface can displace the upper limit of permafrost in the soil and therefore ALT, but be too small to completely thaw the permafrost within the soil.

We are interested in how the modifications to the bottom boundary produce changes in the carbon pools of the permafrost region, and how the areal extent of the permafrost region evolves in time. To avoid ambiguities, we define the region of study as the region of the Northern Hemisphere where near-surface permafrost is present at the initial time of the simulations in 1901 CE (Fig. 4). This region covers parts of Northern Canada, Alaska, Siberia, Tibet, Inner Scandinavia, and the coast of Greenland. The interior of Greenland, covered by glaciers, is represented in CLM4.5 as a column of ice of thickness 42 m to simulate the surface mass balance of the glacier and pass this information to the land-ice model of CESM1.2, but it does not represent the soil below the glacier.

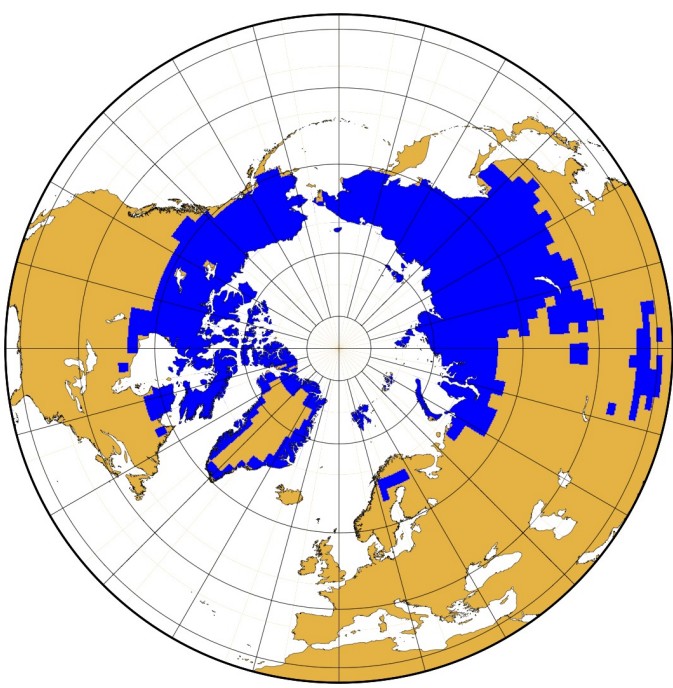

**Figure 4.** Region of study (blue), which corresponds to the extent of near-surface permafrost in the Northern Hemisphere in the year 1901, for the original CLM4.5 model.

## 3.5 Simulations

### 3.5.1 Initialization of the model

We follow the standard spinup procedure (Kluzek, 2013), where the model is initialized with arbitrary pre-initial conditions (no vegetation and uniform subsurface temperature) and driven by a spinup simulation to a steady state (vegetated world adapted to the atmospheric forcings), which are used as initial condition for the simulation. The spinup period required for the initialization of the model depends on the carbon component used by the LSM. In the case of the CLM4.5-BGC, the spinup runs 1000 yr with accelerated decomposition rates (which reduces computational costs and performs consistently well (Thornton and Rosenbloom, 2005)) followed by at least 200 yr with normal decomposition rates. During the spinup phase, we use atmospheric forcings (described in section 3.5.2) that correspond to those of the initial years of the simulation, 1901 to 1910.

Increasing the depth $d$ of the bottom boundary introduces an additional difficulty to the spinup of the model. In the standard spinup procedure, every soil layer is initialized with a temperature of 274 K independent of the grid cell location, then reaches the steady state determined by the local surface boundary conditions during the spinup. For a subsurface thickness of $d = 42.1$ m, 1200 yr of spinup are enough for the subsurface to adapt to the steady state. However, the time needed for the subsurface to reach the steady state is proportional to $d^2$, and 1200 yr is insufficient for the thickest subsurface models. Lengthening the

spinup time for each model of thickness $d$ would make computational costs prohibitive. To avoid this problem, we only use the standard spinup procedure for the model with the original bottom boundary depth, $d = 42.1$ m. The initial conditions for the models with $d > 42.1$ m are obtained by downward continuing the temperature of the 15th layer with the geothermal gradient used as bottom boundary condition ($0$ mW m$^{-2}$ for our experiments with modified $d$). This approach is possible because there are no other variables than temperature in bedrock layers, such as water or carbon content. In addition, as these models start from a common initial state, we can determine any difference in the final state as due to the parameter $d$ exclusively.

### 3.5.2 Simulation of the 1901-2300 period

Each version of the LSM is run offline between 1901 CE and 2300 CE, taking prescribed atmospheric variables from external sources as input to force the model. These simulations include two phases depending on the input used, (1) between 1901-2005, from reanalysis of historical data, and (2) between 2006-2300, from the IPCC climate projection under two warming scenarios (Thomson et al., 2011; Riahi et al., 2011).

The first phase is a historical 20th century simulation between 1901-2005. The forcing data are taken from the CRUNCEP dataset (Viovy, 2018), combination of the Climate Research Unit Time-Series (CRU-TS) monthly climatology (Harris et al., 2014) and the National Centers for Environmental Prediction (NCEP) reanalysis (Kalnay et al., 1996) between the years 1901 and 2005.

The second phase continues the first phase between 2006-2300, forces the LSM with the atmospheric output from a simulation for a specific trajectory of greenhouse gas concentration. These trajectories, called Representative Concentration Pathways (RCPs), are based on scenarios of future human emissions and provide a basis to the climate research community for modeling experiments in the long and short terms (Van Vuuren et al., 2011).

We use two scenarios, RCP 4.5 and RCP 8.5, for our simulations after 2005. RCP 4.5 is an mitigation scenario of anthropogenic emissions where radiative forcing reaches $4.5$ W m$^{-2}$ in 2100 (Thomson et al., 2011). In comparison, RCP 8.5 is a high emissions scenario of considerable increase of greenhouse gas emissions and concentrations, leading to a radiative forcing of $8.5$ W m$^{-2}$ at the end of the 21st century (Riahi et al., 2011).

Forcing datasets of monthly averages are provided by the Earth System Grid (Stern, 2013) for both scenarios. To produce 6h-resolution datasets suitable for CLM4.5, we calculated the 6h-anomalies to monthly average for temperature and precipitation in the years 1996-2005 of the CRUNCEP dataset, and added this 10 yr series of anomalies to the monthly datasets cyclically, starting in 2006. The 6h-resolution datasets produced this way were then used to force the land system between 2006-2300 for the two scenarios. The mean SAT over the land area for the duration of our simulation time is shown in Fig. 5. The mean SAT in the last decade 2290-2300 is $\approx 2$ K higher than in the decade 2000-2010 for the RCP 4.5 scenario, while for the RCP 8.5 scenario temperature rises $\approx 9.5$ K.

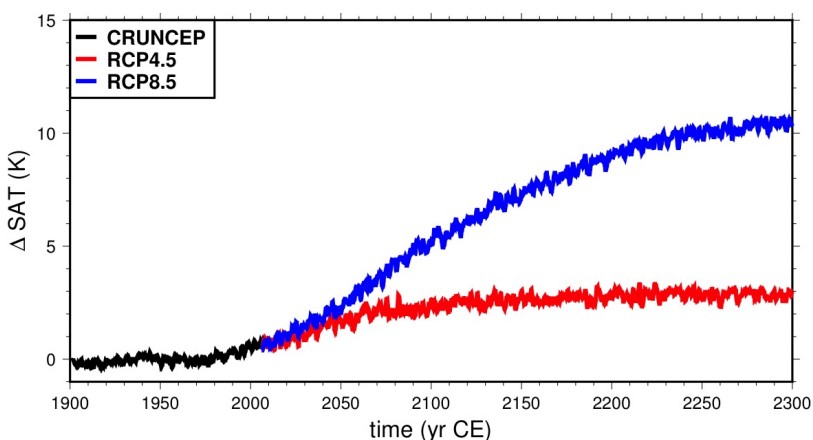

**Figure 5.** Mean SAT over land relative to the 20th century mean, from the CRUNCEP dataset (black) and the RCP 4.5 (red) and RCP 8.5 (blue) scenarios. Data taken from Viovy (2018); Thomson et al. (2011); Riahi et al. (2011).

## 4 Results

### 4.1 Heat storage

#### 4.1.1 Effect of the depth of the bottom boundary

**Table 1.** Heat stored in the subsurface since 1901 CE at the years 2000, 2100, 2200 and 2300 CE for the RCP 4.5 and RCP 8.5 scenarios.

| | | RCP 4.5 | | | RCP 8.5 | | |
|---|---|---|---|---|---|---|---|
| $d$ (m) | ΔH 1901-2000 (ZJ) | ΔH 1901-2100 (ZJ) | ΔH 1901-2200 (ZJ) | ΔH 1901-2300 (ZJ) | ΔH 1901-2100 (ZJ) | ΔH 1901-2200 (ZJ) | ΔH 1901-2300 (ZJ) |
| 3.8 | 2.20 | 4.99 | 4.86 | 4.83 | 6.60 | 9.66 | 10.89 |
| 42.1 | 6.03 | 24.14 | 26.91 | 27.74 | 44.41 | 78.13 | 92.64 |
| 92.1 | 7.31 | 41.12 | 53.91 | 57.84 | 69.90 | 148.01 | 191.37 |
| 142.1 | 7.63 | 45.96 | 69.59 | 81.52 | 75.65 | 178.98 | 255.66 |
| 192.1 | 7.66 | 46.81 | 75.02 | 93.67 | 76.59 | 187.63 | 282.66 |
| 242.1 | 7.66 | 46.94 | 76.35 | 98.15 | 76.73 | 189.52 | 291.36 |
| 292.1 | 7.66 | 46.95 | 76.67 | 99.60 | 76.74 | 189.89 | 293.77 |
| 342.1 | 7.66 | 46.96 | 76.75 | 100.00 | 76.72 | 189.92 | 294.31 |

The results summarized in Table 1 confirm the calculations of the absorption of heat by the subsurface discussed in section 2. The heat absorbed by the subsurface varies with time between models of different subsurface thickness $d$ (Fig. 6). If the

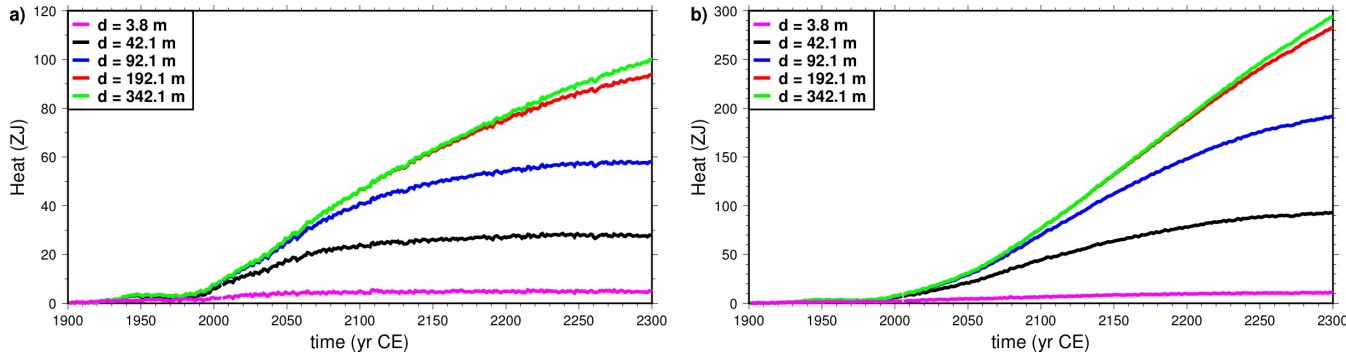

**Figure 6.** Heat stored in the subsurface as function of time, for models of subsurface thickness $d$ of 3.8 m (magenta), 42.1 m (black), 92.1 m (blue) 192.1 m (red) and 342.1 m (green). a) Simulations forced with CRUNCEP + RCP 4.5 data. b) Simulations forced with CRUNCEP + RCP 8.5 data. Note the scale difference between scenarios RCP 4.5 and RCP 8.5.

bottom boundary is too shallow, the thermal signal from the surface reaches the bottom boundary and further absorption of heat is hindered. For the original depth of the CLM4.5, $d = 42.1$ m, after 100 yr its subsurface absorbs considerably less heat than for the deeper models. As we progressively increase the thickness of the subsurface, this effect is reduced and delayed. By the end of the simulation, the thickest model ($d = 342.1$ m) has absorbed 72 ZJ ($72 \times 10^{21}$ J) in the RCP 4.5 scenario and

5    201 ZJ in the RCP 8.5 scenario, which are respectively 3.6 and 3.17 times the heat stored by the original model. If compared to the thinnest model (3.8 m) instead, the thickest model absorbs 20 and 27 times more heat.

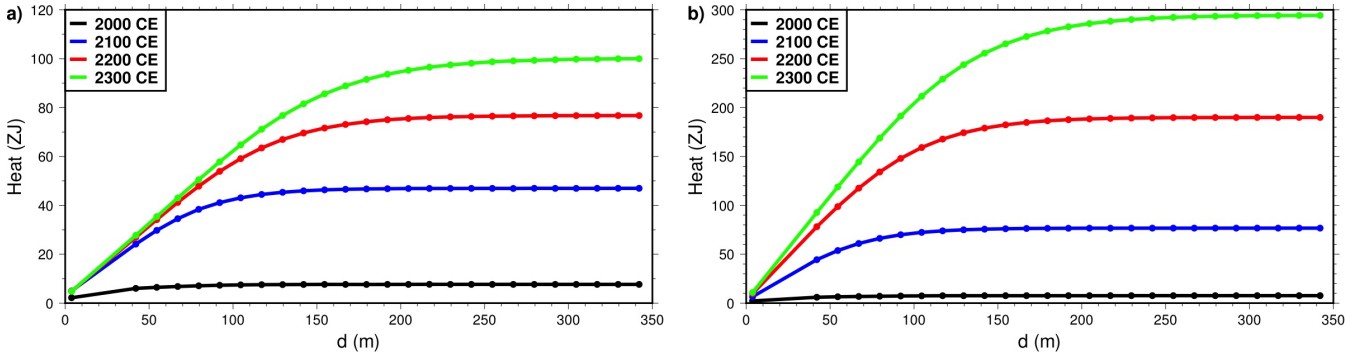

**Figure 7.** Heat stored in the subsurface as function of subsurface thickness, at the years 2000 (black), 2100 (blue), 2200 (red) and 2300 (green). a) Simulations forced with CRUNCEP + RCP 4.5 data. b) Simulations forced with CRUNCEP + RCP 8.5 data. Note the vertical scale difference between the two panels.

At a given time, the heat absorbed by the subsurface increases with the depth of the bottom boundary $d$ of the model (Fig. 7). The amount of heat is not proportional to $d$ and levels off when $d$ increases past a specific threshold. This value is the thickness required by the model to keep the heat absorbed close to the maximum absorbed by the half space. For a threshold of 95%,

10    this depth is $\approx 90$ m if the simulation runs for 100 yr (until 2000 CE). If we look at the heat absorbed after 400 yr, this depth is $\approx 200$ m in the RCP 4.5 scenario (Fig. 7a), and $\approx 180$ m in the RCP 8.5 scenario (Fig. 7b), which confirms the theoretical

estimates. This difference shows that the SAT forcing, dependent on the scenario, has only a small influence on the threshold. It is determined by the heat conduction time across a layer of thickness $d$, that is the relationship $d \propto \sqrt{\kappa t}$ deduced from Eq. (5).

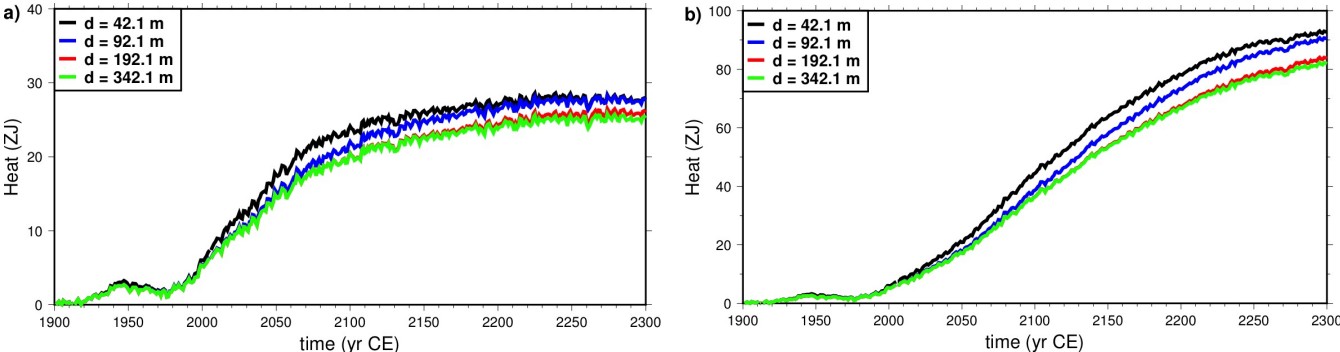

**Figure 8.** Heat stored in the upper 42.1 m as function of time, for models of subsurface thickness $d$ of 42.1 m (black), 92.1 m (blue) 192.1 m (red) and 342.1 m (green). a) Simulations forced with CRUNCEP + RCP 4.5 data. b) Simulations forced with CRUNCEP + RCP 8.5 data. Note the vertical scale difference between the two panels.

Deepening the bottom boundary below 42.1 m also affects the storage of heat within the layers above (Fig. 8). The thermal
5  signal is reflected by the bottom boundary, further heating the region above, but as we increase $d$, this additional heat decreases. For the thickest model ($d = 342.1$ m), the upper 42.1 m of the subsurface gain 2.5 ZJ less than the original model in the RCP4.5 scenario (Fig. 8a) and 10.7 ZJ in the RCP8.5 scenario (Fig. 8b), which correspond respectively to a decrease of 9% and of 11.6%.

**Table 2.** Heat stored in the soil (upper 3.8 m) since 1901 CE at the years 2000 and 2300 CE for the RCP 4.5 and RCP 8.5 scenarios, as function of subsurface thickness $d$.

|  |  |  | RCP 4.5 | | RCP 8.5 | |
| --- | --- | --- | --- | --- | --- | --- |
| $d$ (m) | $\Delta H$ 1901-2000 (ZJ) | $\Delta H(d)/\Delta H(42.1)$ 1901-2000 | $\Delta H$ 1901-2300 (ZJ) | $\Delta H(d)/\Delta H(42.1)$ 1901-2300 | $\Delta H$ 1901-2300 (ZJ) | $\Delta H(d)/\Delta H(42.1)$ 1901-2300 |
| 3.8 | 2.201 | 1.2021 | 4.833 | 1.0805 | 10.889 | 1.0339 |
| 42.1 | 1.831 | 1 | 4.473 | 1 | 10.532 | 1 |
| 92.1 | 1.816 | 0.9917 | 4.465 | 0.9984 | 10.471 | 0.9942 |
| 142.1 | 1.813 | 0.9904 | 4.441 | 0.9930 | 10.392 | 0.9867 |
| 192.1 | 1.817 | 0.9926 | 4.427 | 0.9898 | 10.348 | 0.9826 |
| 242.1 | 1.813 | 0.9904 | 4.419 | 0.9879 | 10.330 | 0.9809 |
| 292.1 | 1.810 | 0.9885 | 4.412 | 0.9864 | 10.332 | 0.9810 |
| 342.1 | 1.814 | 0.9908 | 4.411 | 0.9863 | 10.321 | 0.9800 |

Most of the subsurface is considered as bedrock, where the only heat transport process is thermal diffusion. The region of most interest is the soil, (upper 3.8m) where biogeochemical processes, sensitive to temperature, take place. The heat absorbed by the soil has been summarized in Table 2. The heat absorbed by the soil is overestimated for the shallow bottom boundary variants of the model in the same manner as it was for the upper 42.1 m, but this effect is much smaller.

The quantitative differences in Table 2 are small and better analyzed as the heat gained by the soil in each model as relative to the heat gained in the original model (42.1 m thick). Compared to the original model, the heat stored in the deepest models is $\approx 1\%$ less after 100 years of simulation, and $\approx 1.3\%$ at the end of the RCP 4.5 scenario and $\approx 2\%$ at the end of the RCP 8.5 scenario. The thinnest model (3.8 m) stores 20% more heat than the original model after 100 yr, a relative difference that is reduced to 8% (RCP 4.5) and 3.4% (RCP 8.5) by the end of the simulations, which shows that decreasing the thickness of

the subsurface produces a larger effect on heat storage than increasing it. The differences between scenarios RCP 4.5 and RCP 8.5 are caused by the yearly changes of SAT forcing (Fig. 5), which increases at the fastest rate during the 21st century in both RCP scenarios.

### 4.1.2    Effect of the bottom heat flux

In a purely conductive thermal regime of the subsurface, the value of the heat flux used as bottom boundary condition does

not affect heat diffusion. This is not the case for the soil, because in CLM4.5 the thermal properties of the soil depend on temperature through the water/ice content. However, because of the shallowness of the soil, the geothermal gradient does not raise soil temperature sufficiently to affect heat propagation. Therefore, while the heat content of the subsurface increases with the lower boundary heat flux, it does not affect its time evolution.

For the 42.1 m of the subsurface, the bottom heat flux increases the heat content by $2.058 \pm 0.006$ ZJ for each 20 mW m$^{-2}$.

This offset is independent of the forcing scenario and constant in time. The soil (upper 3.8 m) exhibits the same behavior as for the upper 42.1 m but with smaller amplitude, where heat content is offset by $0.043 \pm 0.004$ ZJ for every 20 mW m$^{-2}$ increase, regardless of the scenario.

This increase of soil heat content due to the bottom heat flux does not translate into a uniform increase of soil temperature across individual cells, because soil composition and thermal properties vary. Each 20 mW m$^{-2}$ increase of bottom heat flux

increases the temperature of the deepest soil layer (node at depth 2.86 m) by $0.04 \pm 0.01$ K. Using the mean continental heat flux value of 60 mW m$^{-2}$ as bottom boundary condition increases the temperature of the bottom soil layer by $0.12 \pm 0.03$ K and that of the bottom bedrock layer (node depth at 35.1 m) by $0.8 \pm 0.04$ K.

### 4.2    Permafrost

### 4.2.1    Intermediate-depth Permafrost

Given the increasing SAT anomalies used to force the model (Fig. 5), we expect to observe a continuous decrease in the area extent of permafrost during the simulation period. The SAT warming signal is expected to propagate downward and, for a shallow bottom boundary, to be reflected back to the surface, thus overheating the subsurface. A deeper bottom boundary

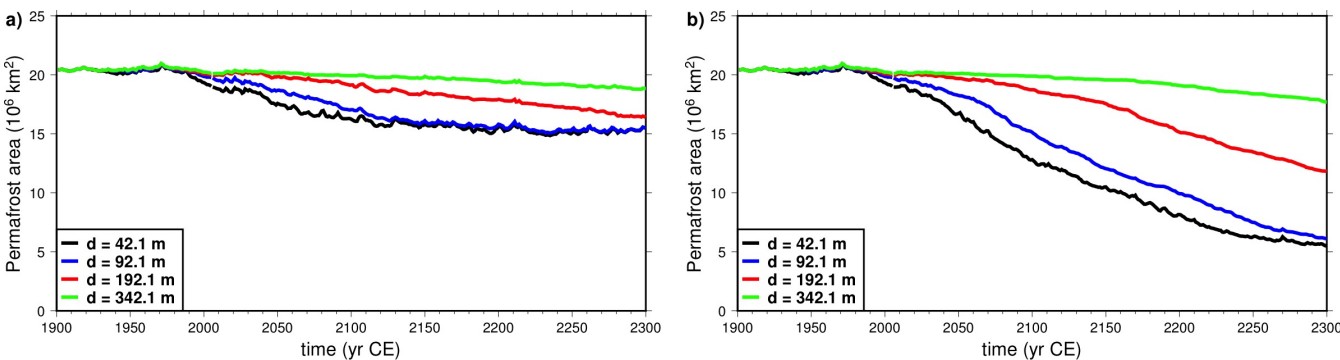

**Figure 9.** Northern Hemisphere intermediate-depth (0-42.1 m) permafrost area as function of time. Model versions with bottom boundary depth $d$ at 42.1 m (black), 92.1 m (blue) 192.1 m (red) and 342.1 m (green). a) Simulations forced with CRUNCEP + RCP 4.5 data. b) Simulations forced with CRUNCEP + RCP 8.5 data.

attenuates this effect and therefore decreases the rate of permafrost thawing. Because a shallow lower boundary heats the subsurface from the bottom, this overheating is highest at depth, and the effect on the soil is less noticeable.

In our simulations, the area with intermediate-depth permafrost in the Northern Hemisphere (Fig. 9) has an initial areal extent of $20.4 \times 10^6$ km$^2$ in 1901. At the end of the RCP 4.5 scenario, this area has been reduced by $4.94 \times 10^6$ km$^2$ (24.1% of the initial area) for the original model and by $1.59 \times 10^6$ km$^2$ (7.8%) for the thickest model. For the RCP 8.5 scenario, the area losses of intermediate-depth permafrost are $14.85 \times 10^6$ km$^2$ (72.7%) for the original model and $2.74 \times 10^6$ km$^2$ (13.4%) for the thickest model.

For both scenarios, the decrease of intermediate-depth permafrost area becomes smaller as we increase the depth of the bottom boundary (Fig. 9). Each increase of the thickness of the subsurface produces diminishing returns, reaching a plateau where the permafrost area is not affected by a further increase of the bottom boundary depth. The depth at which this plateau is reached increases with the length of the simulation, and by the end of the simulations at 2300, it exceeds the largest bottom boundary depth (342.1 m) used in our versions of the model. Table 3 summarizes the evolution of intermediate-depth permafrost for the original CLM4.5 and the modified versions of $d = 342.1$ m and $F_B = 80$ mW m$^{-2}$.

**Table 3.** Areal extent of intermediate-depth permafrost at 1901 CE, 2000 CE and 2300 CE for the RCP 4.5 and RCP 8.5 scenarios.

| Subsurface parameters | | CRU-NCEP | | RCP 4.5 | | RCP 8.5 | |
|---|---|---|---|---|---|---|---|
| $d$ (m) | $F_B$ (mW m$^{-2}$) | PF area 1901 ($\times 10^6$ km$^2$) | PF area 2000 ($\times 10^6$ km$^2$) | PF area 2300 ($\times 10^6$ km$^2$) | Fraction PF lost 1901-2300 (%) | PF area 2300 ($\times 10^6$ km$^2$) | Fraction PF lost 1901-2300 (%) |
| 42.1 | 0 | 20.43 | 19.33 | 15.49 | 24.18 | 5.58 | 72.68 |
| 42.1 | 80 | 19.85 | 18.65 | 14.72 | 25.84 | 5.11 | 74.25 |
| 342.1 | 0 | 20.43 | 20.21 | 18.84 | 7.78 | 17.69 | 13.41 |

The addition of a non-zero heat flux boundary condition at the LSM's bottom boundary has a small effect on intermediate-depth permafrost area (Table 3). The initial extent of intermediate-depth permafrost is reduced by $0.15 \pm 0.07 \times 10^6 \ \mathrm{km^2}$ (0.7%) for every increase of $20 \ \mathrm{mW \ m^{-2}}$ in $F_B$. This difference does not remain constant during the simulation, each increase $20 \ \mathrm{mW \ m^{-2}}$ of $F_B$ reduces the intermediate-depth permafrost area at the end of the simulation by $0.19 \pm 0.14 \times 10^6 \ \mathrm{km^2}$ in the RCP 4.5 scenario and by $0.12 \pm 0.05 \times 10^6 \ \mathrm{km^2}$ in the RCP 8.5 scenario, a decrease relative to the initial permafrost extent of 2.1% and 1.2% respectively.

### 4.2.2 Near-surface permafrost

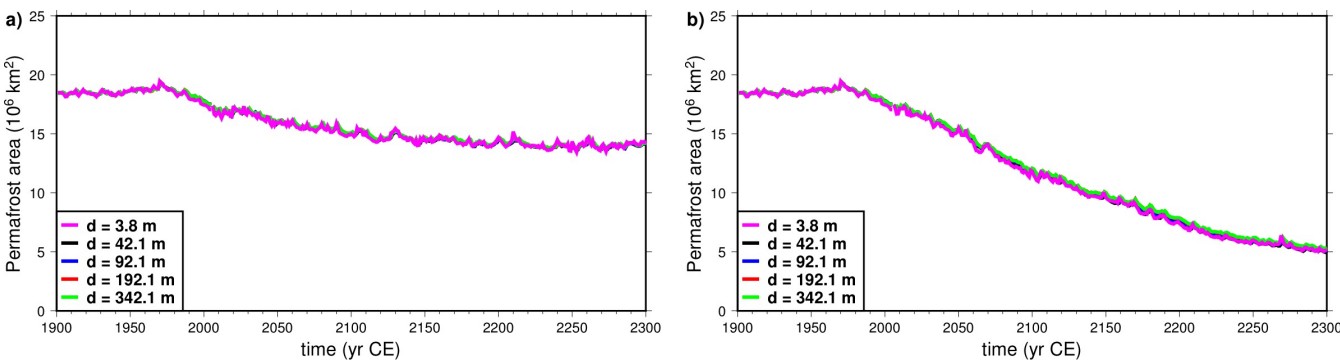

**Figure 10.** Northern Hemisphere near-surface permafrost area as function of time. Model versions with bottom boundary depth at 3.8 m (magenta), 42.1 m (black), 92.1 m (blue) 192.1 m (red) and 342.1 m (green). a) Simulations forced with CRUNCEP + RCP 4.5 data. b) Simulations forced with CRUNCEP + RCP 8.5 data.

The near-surface permafrost (within the upper 3.8 m) area in the Northern Hemisphere is much less affected by the thickness of the model than the intermediate-depth permafrost (Fig. 10). The initial extent of near-surface permafrost is $18.45 \times 10^6 \ \mathrm{km^2}$, and by 2300 under the RCP 4.5, this area has been reduced by $4.27 \times 10^6 \ \mathrm{km^2}$ (23.1%) for the original model and $4.20 \times 10^6 \ \mathrm{km^2}$ (22.7%) for the thickest model, a relative difference of 1.8%. In the RCP 8.5 case, the permafrost area is reduced by $13.37 \times 10^6 \ \mathrm{km^2}$ (72.5%) for the original model and $13.11 \times 10^6 \ \mathrm{km^2}$ (71.1%) for the thickest model, an area decrease 1.9% smaller. Reducing the thickness of the model to 3.8 m only produces differences in the order of 0.5-1.1% for the areal extent of near-surface permafrost.

The effect of the bottom heat flux $F_B$ on near-surface permafrost area is similar to that on intermediate-depth permafrost, but quantitatively smaller (Table 4). Each $20 \ \mathrm{mW \ m^{-2}}$ increase reduces the initial near-surface permafrost extent by $0.05 \pm 0.04 \times 10^6 \ \mathrm{km^2}$ (0.3%). At 2300, this increase in bottom heat flux reduces the final permafrost extent by $0.09 \pm 0.08 \times 10^6 \ \mathrm{km^2}$ (0.6%) for the RCP 4.5 scenario and by $0.04 \pm 0.01 \times 10^6 \ \mathrm{km^2}$ (0.8%) for the RCP 8.5 scenario. The results for the original CLM4.5 and the modified versions of $d = 3.8$ m, $d = 342.1$ m and $F_B = 80 \ \mathrm{mW \ m^{-2}}$ are summarized in Table 4.

The initial state of the subsurface in 1901 is identical for model versions with different subsurface thickness, provided they use the same bottom heat flux. The temperature of the upper subsurface increases at a slower rate for a deeper bottom boundary,

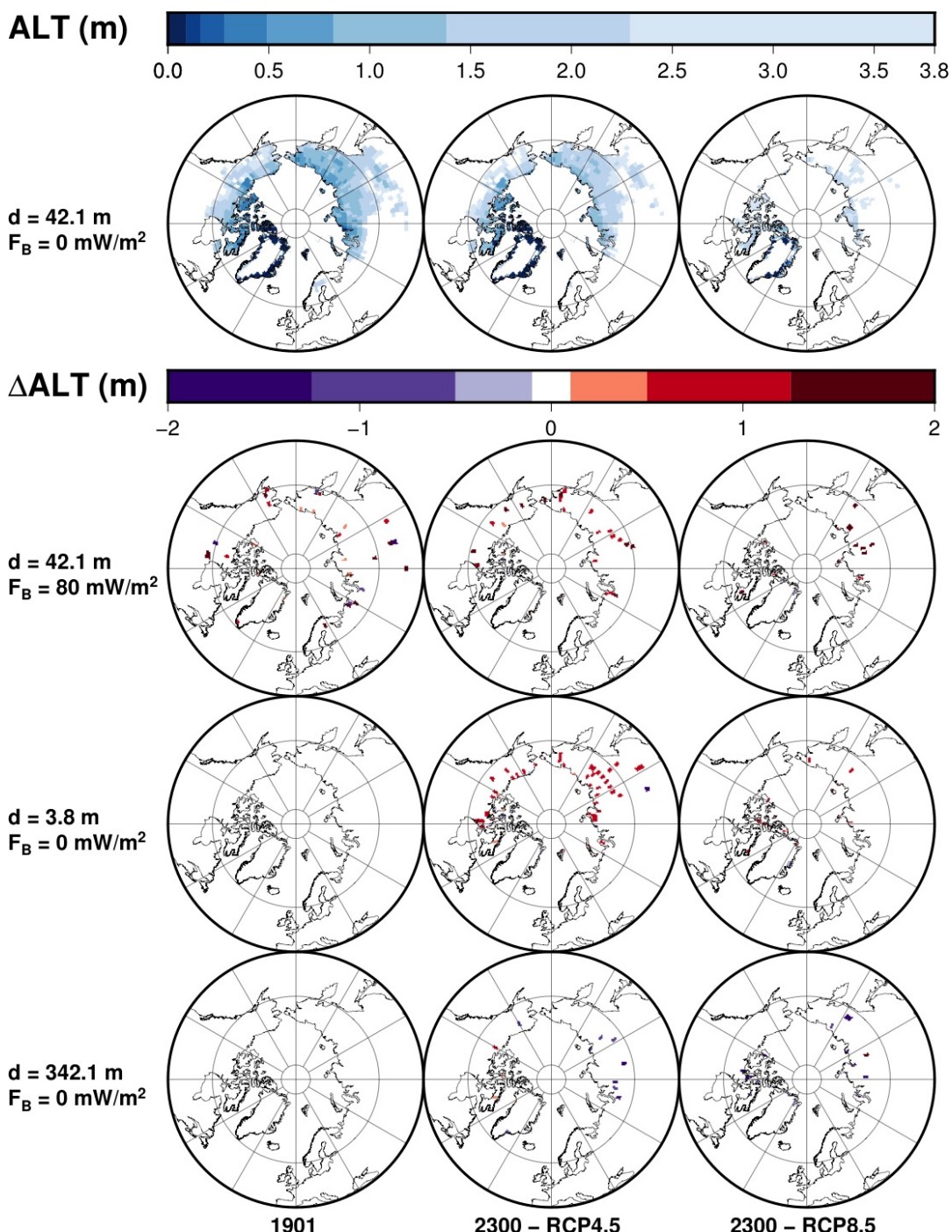

**Figure 11.** Active Layer Thickness for the unmodified model (top row), and differences to the original model at each time frame for the modified model with $80 \, \mathrm{mW \, m^{-2}}$ (second row), the modified model with $d = 3.8 \, \mathrm{m}$ (third row) and the modified model with $d = 342.1 \, \mathrm{m}$ (bottom row). Time frames at 1901 CE and 2300 CE for the scenarios RCP 4.5 and RCP 8.5.

**Table 4.** Areal extent of near-surface permafrost at 1901 CE, 2000 CE and 2300 CE for the RCP 4.5 and RCP 8.5 scenarios.

| Subsurface parameters | | CRU-NCEP | | RCP 4.5 | | RCP 8.5 | |
|---|---|---|---|---|---|---|---|
| $d$ (m) | $F_B$ (mW m$^{-2}$) | PF area 1901 ($\times 10^6$ km$^2$) | PF area 2000 ($\times 10^6$ km$^2$) | PF area 2300 ($\times 10^6$ km$^2$) | Fraction PF lost 1901-2300 (%) | PF area 2300 ($\times 10^6$ km$^2$) | Fraction PF lost 1901-2300 (%) |
| 3.8 | 0 | 18.45 | 17.40 | 14.13 | 23.41 | 5.12 | 72.25 |
| 42.1 | 0 | 18.45 | 17.80 | 14.17 | 23.17 | 5.07 | 72.49 |
| 42.1 | 80 | 18.25 | 17.09 | 13.80 | 24.40 | 4.90 | 73.15 |
| 342.1 | 0 | 18.45 | 17.76 | 14.25 | 22.75 | 5.34 | 71.07 |

thus the ALT increases at a slower rate for model versions with deeper subsurface. At the end of the simulations in 2300, the ALT is in some areas larger for the original model (42.1 m) than for the model with thickness increased to 342.1 m and smaller than for the model with thickness of 3.8 m, for both scenarios (Fig. 11).

The bottom heat flux increases temperature proportionally to the flux and the depth. Therefore, bottom heat flux does not
5  alter ALT if permafrost is shallow. Where ALT is large, the higher temperature due to the bottom heat flux is enough to induce thawing and lower the upper limit of permafrost (Fig. 11).

### 4.3 Carbon

#### 4.3.1 Soil Carbon

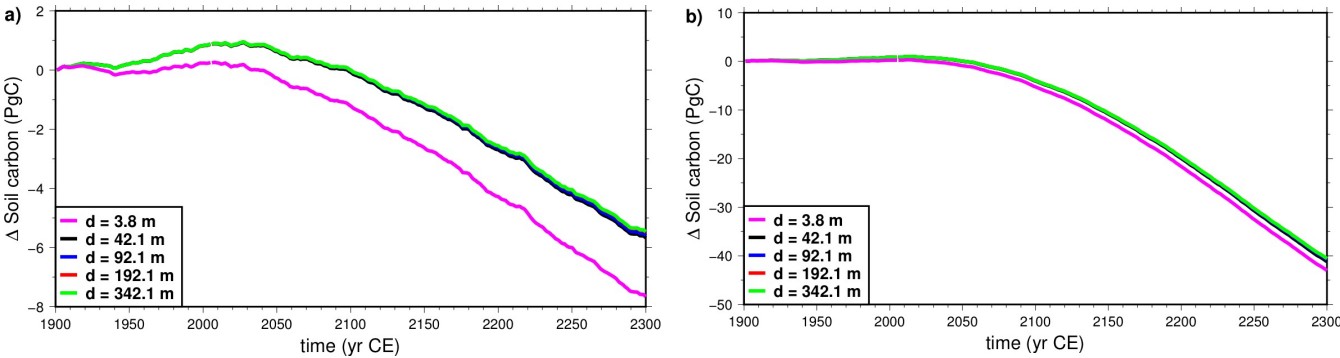

**Figure 12.** Evolution of soil carbon pool in the Northern Hemisphere permafrost region, compared to the size for the original model at 1901 CE. Models with varying bottom boundary depth. a) Simulations forced with CRUNCEP + RCP 4.5 data. b) Simulations forced with CRUNCEP + RCP 8.5 data. Note the vertical scale difference between the two panels.

The size of the soil carbon pool increases during the first $\approx 150$ yr of simulation and thereafter begins decreasing, losing
10  during the period 1901-2300 a total of 5.6 PgC in the RCP 4.5 scenario and 41.2 PgC in the RCP 8.5 scenario, for the original

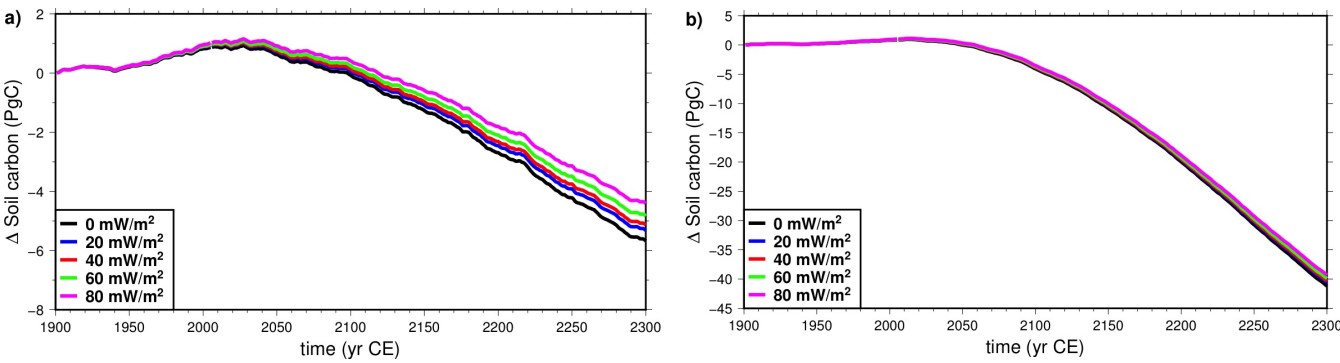

**Figure 13.** Evolution of soil carbon pool in the Northern Hemisphere permafrost region, compared to the size for the original model at 1901 CE. Models with varying basal heat flux. a) Simulations forced with CRUNCEP + RCP 4.5 data. b) Simulations forced with CRUNCEP + RCP 8.5 data. Note the vertical scale difference between the two panels.

model. Increasing the depth of the bottom boundary reduces the loss of soil carbon, as expected because it slows the rate of permafrost thawing. The loss of soil carbon for the thickest subsurface (342.1 m) is 0.15 PgC (3.6%) less than for the original subsurface model (42.1 m) in the RCP 4.5 scenario, and 0.56 PgC (1.3%) less in the RCP 8.5 scenario (Fig. 12). Decreasing the subsurface thickness to 3.8 m produces a much larger effect, with soil carbon decreasing by 7.66 PgC (RCP 4.5) and 43.02

PgC (RCP 8.5) during the simulation, which amounts to an increase of the soil carbon lost in the period 1901-2300 of 35% (RCP 4.5) and 4.4% (RCP 8.5), relative to the original model.

Increasing the bottom heat flux $F_B$ slows down the rate at which soil carbon in the permafrost region decreases during the simulation. An increase of $20 \text{ mW m}^{-2}$ reduces the loss of soil carbon between 1901 and 2300 by $0.3 \pm 0.1$ PgC (5.6% of the decrease of soil carbon in this period for the original CLM4.5) in the RCP 4.5 scenario and $0.45 \pm 0.2$ PgC (1.1%) in the RCP

8.5 scenario (Fig. 13).

Because the changes in soil carbon due to the modification of model thickness and bottom heat flux are very small relative to the size of the pool, we have calculated the difference in soil carbon between the original model and the modified models with increased thickness $d = 342.1$ m and with bottom heat flux $80 \text{ mW m}^{-2}$. For the original model, the biggest concentrations of soil carbon are located in the permafrost regions of the northern hemisphere, mainly in Alaska and Eastern Siberia (Fig. 14).

While model versions of different thickness share a common initial state, a thicker model increases soil carbon concentration across the region.

Models with different bottom heat flux $F_B$ depart from different initial conditions (since the bottom heat flux determines the thermal steady state of the subsurface). A higher $F_B$ decreases the initial concentration of soil carbon in some areas but increases it in others. These differences can be of the same order of magnitude as the carbon concentration in the original model

in token gridcells. Some cells have quantities of soil carbon in the $80 \text{ mW m}^{-2}$ model half of that of the original model, while other have 10 times as much (Fig. 14).

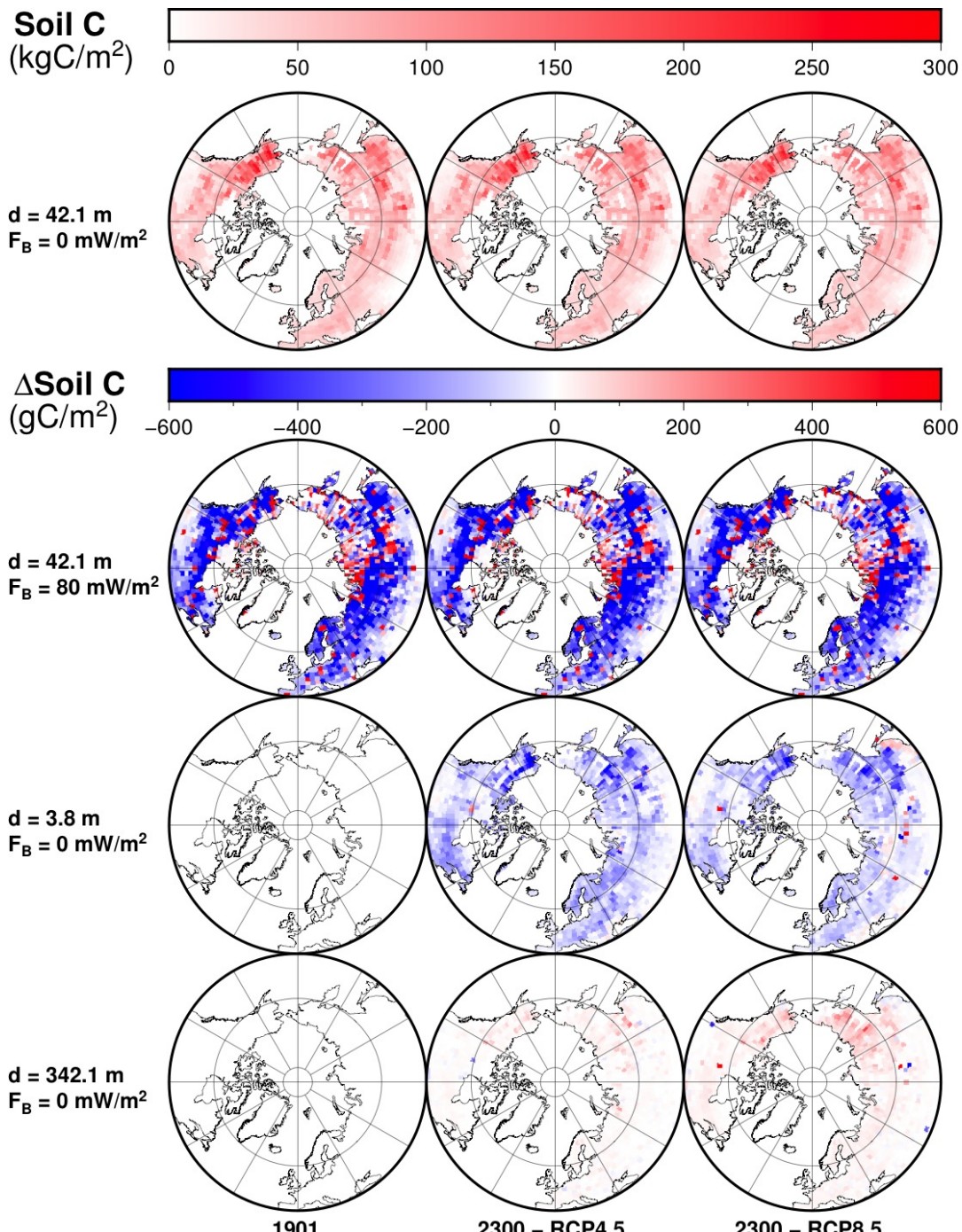

**Figure 14.** Distribution of soil carbon for the original model (top row), and differences to the original model at each time frame for the modified model with $80\ \mathrm{mW\ m^{-2}}$ (second row), the modified model with $d = 3.8$ m (third row) and the modified model with $d = 342.1$ m (bottom row). Time frames at 1901 CE and 2300 CE for the scenarios RCP 4.5 and RCP 8.5.

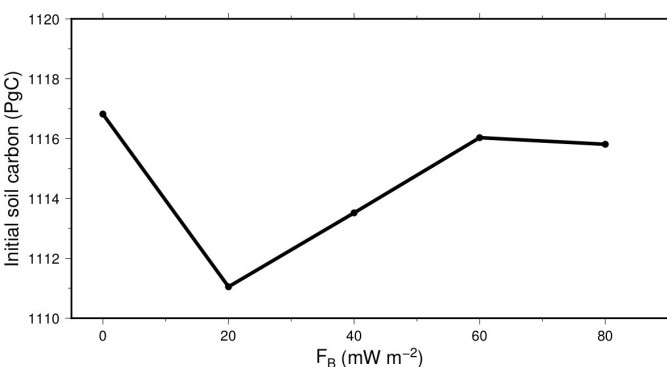

**Figure 15.** Mean initial size (1901-1910) of the soil carbon pool in the Northern Hemisphere permafrost region, in function of bottom heat flux.

Because the local differences on the soil carbon pool due to the bottom heat flux have different signs, the effect on the whole region is proportionally much smaller, and also produces the absence of a consistent trend in the initial size of the soil carbon pool (Fig. 15).

### 4.3.2 Vegetation Carbon

The vegetation carbon in the Northern hemisphere is also affected by the depth of the bottom boundary. Because rising temperatures allow plants to colonize higher latitudes, the vegetation increases for both RCP scenarios, reaching a stable level between 2100-2300. Increasing $d$ and $F_B$ results in more vegetation carbon in some areas and less in others (Fig. 16). For both RCP scenarios, the effect is a net decrease of vegetation carbon in the Northern Hemisphere at the end of the simulations both for greater thickness and for higher bottom heat flux.

While the models with different depth of the bottom boundary $d$ start from the same initial state at 1901, a thicker model leads to slightly smaller masses of vegetation carbon. For the thickest model (342.1 m), the pool of vegetation carbon is $0.17 \pm 0.01$ PgC smaller during the last two centuries of simulation than it is for the original model (42.1 m) in the RCP 4.5 scenario, and $0.11 \pm 0.08$ PgC smaller in the RCP 8.5 scenario. Decreasing the thickness of the model from 42.1 m to 3.8 m produces an effect of comparable magnitude, increasing vegetation carbon by $0.04 \pm 0.01$ PgC for the RCP 4.5 scenario and by $0.08 \pm 0.01$ PgC for the RCP 8.5 scenario during the last two centuries of simulation.

The bottom heat flux also has a small effect in the evolution of vegetation carbon in the Northern Hemisphere for both RCP scenarios. The average vegetation carbon between 2100-2300 for the model with $80 \, \mathrm{mW \, m^{-2}}$ is $0.35 \pm 0.03$ PgC less for the RCP 4.5 scenario and $0.54 \pm 0.05$ PgC less for the RCP 8.5 scenario than for the model with zero basal heat flux, a relative decrease of $0.8 \pm 0.08\%$ and $1.2 \pm 0.1\%$ respectively.

The bottom heat flux changes the initial stable size of the vegetation carbon pool in individual cells, that results in a positive change over the North Hemisphere permafrost region (Fig. 17). There is a consistent linear increase of $0.066 \pm 0.02$ PgC of the initial vegetation for each $20 \, \mathrm{mW \, m^{-2}}$ increase of the bottom heat flux.

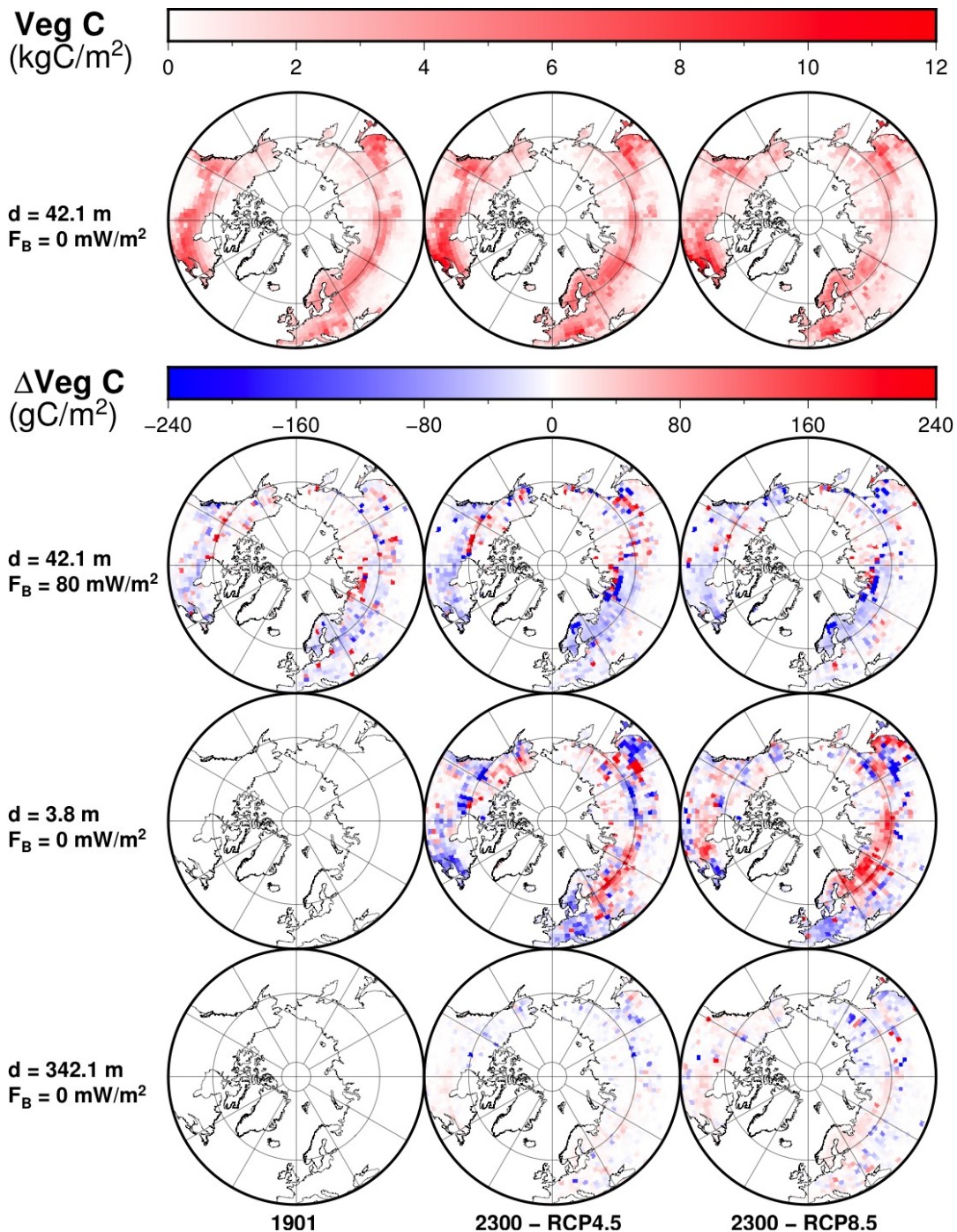

**Figure 16.** Distribution of vegetation carbon for the original model (top row), and differences to the original model at each time frame for the modified model with $80\ \mathrm{mW\ m^{-2}}$ (second row), the modified model with $d = 3.8\ \mathrm{m}$ (third row) and the modified model with $d = 342.1\ \mathrm{m}$ (bottom row). Time frames at 1901 CE and 2300 CE for the scenarios RCP 4.5 and RCP 8.5.

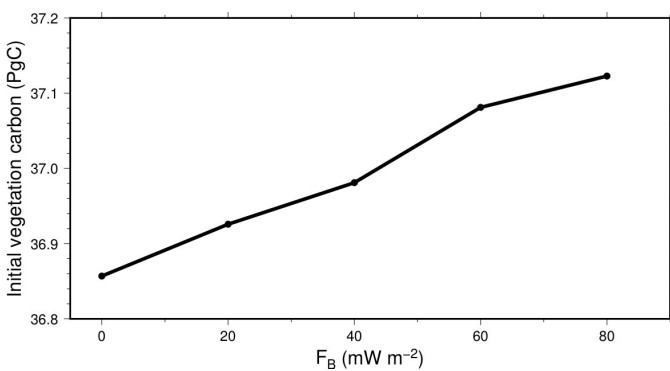

**Figure 17.** Mean initial size (1901-1910) of the vegetation carbon pool in the Northern Hemisphere permafrost region, in function of bottom heat flux.

### 4.3.3 Methane

Methane is produced by methanogenic microbes in the anaerobic fraction of soil. Therefore, it concentrates in areas where the water table rises high enough to reach the carbon-rich soil near the surface, or in inundated areas. The production of methane in natural wetlands is mainly located in the tropical areas, responsible for 64%-88% of the global wetland production (O'Connor
et al., 2010).

In our CLM4.5-BGC simulations, most of the methane production is concentrated in the Northern Hemisphere cold regions, including not only the permafrost region but the areas of seasonal soil freezing as well (Fig. 18). In contrast, the tropical areas produce almost no methane. The reason lies in the unconfined aquifer present below the soil in the hydrological model of the CLM4.5, which allows the subsurface to absorb water before the water table rises to the upper soil layers, where most of the
soil carbon is concentrated. In the simulations, the water table rarely rises above a depth of 3.8 m during the monsoon season. High-latitude areas have low water tables as well, but they get partially inundated during the year because the soil is frozen (impeding the percolation of water), and can produce methane.

In the Northern Hemisphere there are significant differences in the production of methane due to the bottom heat flux and the depth of the bottom boundary. In a few areas, the difference in methane production can be as high as 50-80% with respect
to the original model. However, as the sign of these differences can be either positive or negative, the net effect over methane production is small.

The net effect of the subsurface thickness and the bottom heat flux on the global methane production is much smaller than for the localized areas displayed in Fig. 18. Increasing the thickness of the model from 42.1 m to 342.1 m can result in increases and decreases of global methane production during the simulation between 0.1 to 0.2 $\text{TgC yr}^{-1}$ (1 TgC = $10^{12}$ g of C), only
0.3-0.5% of the methane production at 2300 for the scenarios RCP 4.5 and RCP 8.5, respectively. Decreasing subsurface thickness from 42.1 m to 3.8 m results in methane emissions rising by $1.11 \pm 0.35 \text{ TgC yr}^{-1}$ in the RCP 4.5 scenario and by $0.83 \pm 0.34 \text{ TgC yr}^{-1}$ in the RCP 8.5 scenario, a relative increase of 1.5-2% that leads to a larger depletion of the soil carbon pool in the long term. The bottom heat flux has a slightly larger effect, as a bottom heat flux of $80 \text{ mW m}^{-2}$ decreases methane

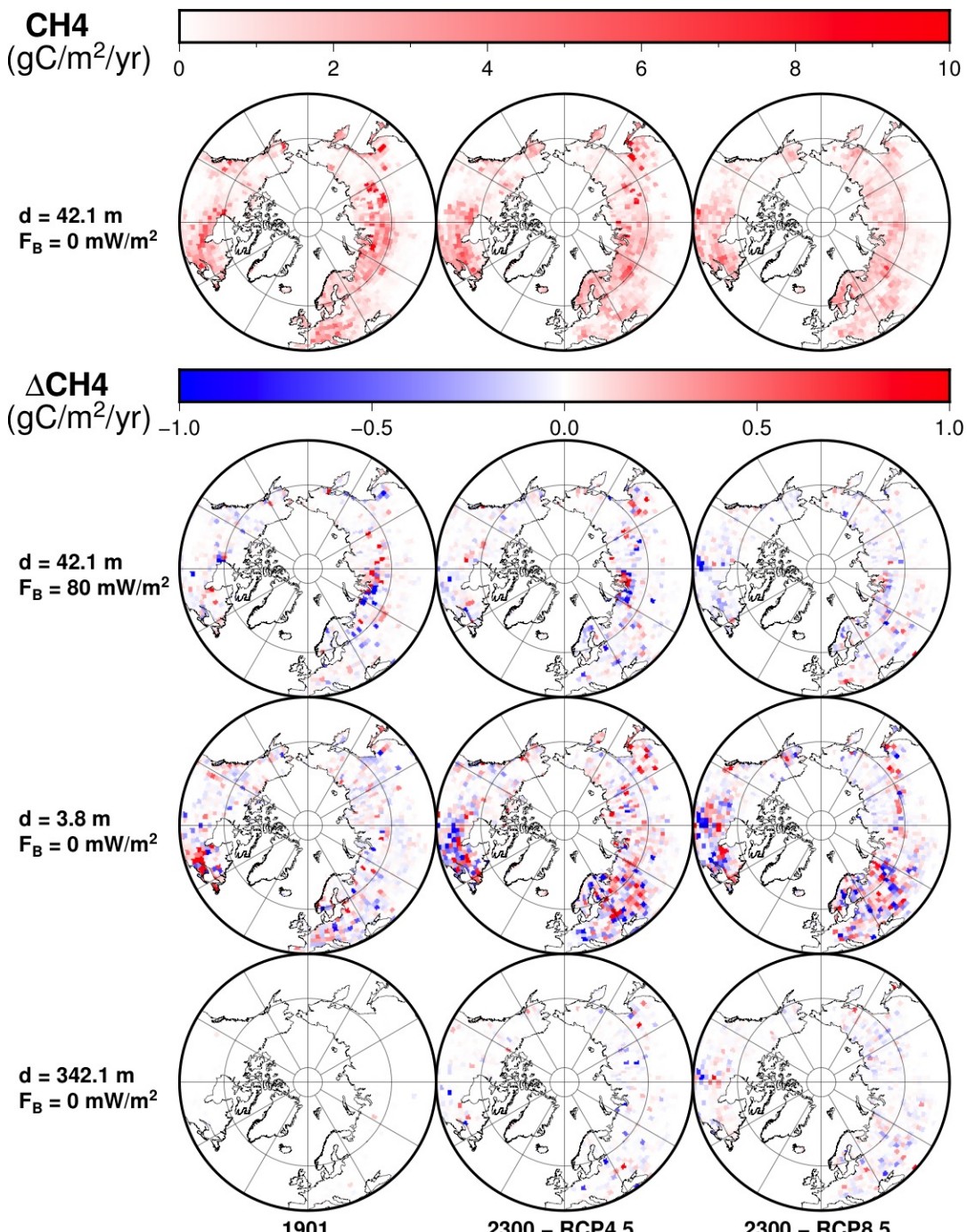

**Figure 18.** Distribution of methane yearly production for the original model (top row), and differences to the original model at each time frame for the modified model with $80\,\mathrm{mW\,m^{-2}}$ (second row), the modified model with $d = 3.8\,\mathrm{m}$ (third row) and the modified model with $d = 342.1\,\mathrm{m}$ (bottom row). Time frames at 1901 CE and 2300 CE for the scenarios RCP 4.5 and RCP 8.5.

production by 0.6 to 1.0 TgC yr$^{-1}$, a relative reduction of 1-1.6% of the total production at 2300 for the scenarios RCP 4.5 and RCP 8.5, respectively.

## 5 Discussion

In this paper we have examined the effects of two simplifications made by most ESMs: not taking the geothermal gradient into account, and using an excessively thin subsurface. This paper follows previous estimations (MacDougall et al., 2008, 2010) and quantifies the effects of these simplifications, through the use of numerical simulations with two sets of modified versions of CLM4.5, one where we increase the thickness of the subsurface, and another where we impose a uniform heat flux at the bottom of the land model.

Our results show that deepening the bottom boundary by 300 m increases the heat stored in the subsurface by 72 ZJ and 201 ZJ at the end of the simulations at 2300 CE, which correspond respectively to 260% and 217% of the heat stored by the original model for scenarios RCP 4.5 and RCP 8.5 respectively. Heat absorption within the soil (upper 3.8 m) is reduced by 1-3% depending on the scenario and the length of the simulation. On the other hand, moving the bottom boundary from 42.1 m to 3.8 m increases the heat absorbed by the soil between 1901 and 2000 by 20%, while the heat absorbed by 2300 only increases by 8% (RCP 4.5) and 3.4% (RCP 8.5), because the heat reflected at the bottom boundary has had time to return to the surface and affect soil temperature. Increasing the bottom heat flux by 20 mW m$^{-2}$ raises the temperature at the bottom of the soil (3.8 m deep) by $0.04 \pm 0.01$ K, with some differences between cells due to the variable thermal properties of soil. For the mean continental heat flux value of 60 mW m$^{-2}$ (Jaupart and Mareschal, 2010) the bottom soil temperature is raised by $0.12 \pm 0.03$ K, and the temperature at the base of the model (42.1 m deep) by $0.8 \pm 0.04$ K.

Permafrost is affected by the depth of the bottom boundary, in a degree that depends on the depth to which we consider permafrost, in the same manner as the heat absorption by the subsurface. Permafrost near the surface is only slightly affected, but for intermediate depth permafrost the thickness of the model has a more significant effect. Increasing the thickness of the subsurface from 42.1 m to 342.1 m reduces the area loss of intermediate-depth permafrost by a factor of 3 in the RCP 4.5 scenario and by a factor of 5.5 in the RCP 8.5 scenario (Fig. 9). The effect of the crustal heat flux on permafrost is proportional to the value of the heat flux and the depth of the permafrost, but even a bottom heat flux of 80 mW m$^{-2}$ reduces intermediate-depth permafrost extent by only 1-2%.

Increasing the depth of the bottom boundary leads to less vegetation and more soil carbon in the Northern Hemisphere permafrost region at the end of the simulations, compared to the thinner models. This is to be expected, as the increasing the depth of the subsurface leads to reduced permafrost loss, which opens less area to vegetation and exposes less soil carbon to microbial activity. These effects are small, as the stable vegetation level reached between 2100-2300 in the thickest model is only reduced by 0.8-1.2% compared to the original model, while soil carbon is reduced by 1.3-3.6%. On the other hand, a subsurface of 3.8 m overestimates the soil carbon lost by 2300 considerably, by as much as 35% in the RCP 4.5 scenario, although this is reduced to 4.4% in the RCP 8.5 scenario, where the loss of soil carbon is greater.

A higher basal heat flux has a regionally variable effect across the Northern Hemisphere, increasing soil carbon and vegetation where near-surface permafrost is present, but decreasing both outside of the permafrost region. The loss of soil carbon in the permafrost region is 4-22% smaller with $80 \text{ mW m}^{-2}$ than with zero basal heat flux, while the initial quantities of carbon in individual gridcells vary between half and 10 times as much as for no heat flux. The heat flux also reduces by 0.8-1.2% the stable vegetation level in this region during the last two centuries of the simulation. On the other hand, the bottom heat flux reduces methane production within areas where permafrost is present but increases it where soil only freezes seasonally.

In CLM4.5 subsurface biogeochemistry only takes place within the soil, the upper 3.8 m. For this reason, the small effect of the bottom boundary depth on near-surface permafrost translates into a small effect on the soil carbon and vegetation pools and the methane production. While the same could be expected from the basal heat flux, it has a variable effect across the Northern Hemisphere, specially in the areas where seasonal freezing of the soil occurs, but no soil permafrost is present.

The Community Land Model is based on many simplifying and often erroneous assumptions (uniform soil thickness, uniform granitic composition of the bedrock, no heat flow, etc). While CLM4.5 uses as uniform soil thickness value of 3.8 m, natural soil thickness varies notably, with an estimated global mean of $\approx 13$ m and reaching depths of several hundred meters i some areas (Shangguan et al., 2017). Soil affected by permafrost is therefore much deeper than in CLM4.5, and future models should use realistic values of soil thickness. The results obtained for intermediate-depth permafrost are therefore useful to understand the effects that the thickness of the subsurface and the bottom heat flux would have in a soil of realistic depth. The uniform soil thickness also affects the hydrology model in CLM4.5 which, in addition to the use of a virtual aquifer with a capacity of 5 m, makes the hydrology model unrealistic. The excessive capacity of this aquifer results in the water table rarely rising above 3.8 m depth, much lower than the natural levels of the water table, specially for the tropical wetlands (Fan et al., 2013). The newer version Community Land Model version 5.0 (CLM5) attempts to address the main issues of the hydrology model in CLM4.5 by eliminating the aquifer and including a spatially variable soil thickness within a range of 0.4 m to 8.5 m, which is still below the global average (Swenson and Lawrence, 2015; Brunke et al., 2016). The soil thickness is derived from survey data where typical values of soil thickness are between 7 m and 10 m (Pelletier et al., 2016), although the growing consensus is that regolith thickness varies between 10-40 m (Clair et al., 2015).

The purpose of the present study is not to build an exact model of the subsurface and calculate its response to climate forcing. It would be pointless because the subsurface has not been surveyed with sufficient resolution to construct such a model. The crustal heat flux is not uniform in the continents but its long wavelength variations across stable continental regions are not very large ($\approx 30 \text{ mW m}^{-2}$). Furthermore, most of the permafrost regions of the Northern hemisphere belong to Precambrian Shields where the heat flow is low and stable (Jaupart and Mareschal, 2015). Varying uniformly the value of the bottom heat flow allows us to establish a quantitative relationship between its value and the effects it has on permafrost and biogeochemistry and to bracket the range of responses of the subsurface to the climate forcing. The land models will also have to address several other erroneous assumptions such as a global granitic bedrock and a constant regolith thickness of a few meters. The most common rocks in the crystalline basement are gneiss, not granite, and large areas in the continents are covered by sedimentary rocks with different thermal properties. The usefulness of the present study is to point out the impact of such deficiencies and the need to introduce the thermal gradient in land models.

A thicker soil in some areas implies that the effect of the basal heat flux and the bottom boundary depth are bigger than our estimations, made for a soil depth of 3.8 m. While soil carbon pools in the permafrost region concentrate within the upper 3 m, additional reserves exist below 3 m which contain $\sim 60\%$ as much carbon as the upper 3 m (Hugelius et al., 2014). If included in the model, these reserves would be more severely affected by the depth of the bottom boundary and the bottom heat flux

than the shallow carbon deposits are. An example of this would be the yedoma and frozen thermokarst deposits, which hold an estimated $211 \pm 160$ PgC of carbon in depths up to 50 m in some areas of Siberia and Alaska (Strauss et al., 2013). Our study showed that the thawing of intermediate-depth permafrost (below 42.1 m) is largely overestimated (3-6 times larger) for a subsurface of 42.1 m than a subsurface of 342.1 m. Therefore, the inclusion of deep carbon deposits in LSMs will require the use of an appropriate subsurface thickness ($\sim 200$ m for a 400 yr simulation).

The methane production in CLM4.5-BGC is dependent on the hydrology model used in CLM4.5, which keeps the water table too low in the tropical regions of the Earth where most (64%-88%) wetland methane is produced (O'Connor et al., 2010). The consequence is that no methane is produced in these regions, and all methane is produced in the Northern Hemisphere where frozen soil can be inundated. Compared to the original model, a bottom heat flux of $80 \ \mathrm{mW} \ \mathrm{m}^{-2}$ causes a reduction of 1-1.6% across the whole permafrost region. Deepening the bottom boundary to 342.1 m only induces variations smaller than

0.5%, while moving the bottom boundary from 42.1 m to 3.8 m consistently increases methane emissions by 1.5-2%. However, there can be differences as high as 50-80% with respect to the original model, located in individual cells near the permafrost frontier. The lack of methane production in tropical regions associated to the hydrology should no longer occur in CLM5.0, which uses a more realistic hydrology model than CLM4.5.

The local variability of the results across the Northern Hemisphere permafrost region is difficult to interpret. Increasing the

thickness of the subsurface or the crustal heat flux reduces the size of the carbon pools and the production of methane in some areas, but it increases it in others. There is a possible explanation for the local differences in the production of methane: the increase in ALT allows more methane to be produced if there is still a frozen soil layer beneath, because it restricts the seepage of water and allows the active layer to be inundated, however if the entire soil thaws, the water can percolate to the aquifer and less methane is produced. This might also explain the local differences in the size of the carbon pool, as the differences in

the production of methane accumulate over time. The local differences in vegetation carbon are more difficult to interpret, but the dominant trend is that warmer soil (because of a larger crustal heat flux or a thinner subsurface) results in more vegetation carbon in the coldest areas of the permafrost region, and less vegetation carbon in the periphery. A tentative explanation is that, while a warmer soil favors the colonization of plants, it may result in less available water in areas where additional heat thaws the soil completely and allows water to percolate to the aquifer and slightly reduce the growth of the vegetation.

The depth of the bottom boundary has a considerable effect on the heat absorbed by the subsurface. We have shown that, in a simulation spanning 400 years, the LSM requires a thickness of at least 200 m to correctly estimate the temperature profile. The thickness $d$ needed increases with the length $t$ of the simulation, but this is not prohibitive for simulations running on much longer timescales, because the depth of the bottom boundary follows a square-root relation $d \propto \sqrt{\kappa t}$. This result matches the estimation obtained from the theoretical analysis, which indicates that we can rely on theoretical estimates of the optimal depth

(Stevens et al., 2007), despite the differences between the theoretical approximation and the numerical model, i.e. the thermal

properties of the upper 3.8 m and the thermal signal from the surface. Longer simulations such as the 1000 yr long simulations of the last millennium ensemble (Stocker et al., 2013), require subsurface thicknesses of $\sim 300 - 350$ m. The computational costs associated to each additional layer are almost negligible when compared to the whole LSM, because the only process taking place in bedrock is thermal diffusion. We also used a fixed thickness for the additional layers, but if we keep the original

scheme where layer thickness increase exponentially, it is possible to increase the thickness of the model to hundreds of meters by adding only a few layers. A downside to this exponential scheme is the loss of resolution to determine the depth range of permafrost, however we consider that the exponential scheme is still a good compromise between the resolution of the subsurface model and its computational cost. If a need to increase the resolution of the layer scheme appears, changing the scaling factor $f_S$ in Eq. (7) is a better solution than abandoning the exponential layer thickness scheme currently used.

We have determined that each successive increase of ground thickness provides diminishing returns for subsurface heat storage and permafrost and soil carbon stability. Therefore, it is to be expected that the improvement from increasing the thickness of the model from 42.1 m to 342.1 m is much smaller than that from increasing it from 3.8 m to 42.1 m. This was already investigated by several studies with the CLM3, where the thickness of the subsurface was increased from 3.5 m to more than 30 m, which improved the estimates of the permafrost significantly during the 20th century (Alexeev et al., 2007; Nicolsky

et al., 2007; Lawrence et al., 2008). A depth of 3.8 m is not enough to damp the annual signal of SAT, and the temperature in a layer of this thickness closely follows the SAT. In future scenarios of global warming such as RCP 4.5 and RCP 8.5, where SAT rises by 2 K and 9.5 K respectively between 2000-2300, this largely overestimates the temperature of the subsurface for the model relative to the real world. In the Northern Hemisphere permafrost region, while we do not detect an effect on the total area extent of near-surface permafrost, the thickness and depth of this permafrost are significantly affected. This results in

consistently higher emissions of methane, which leads to considerable overestimates of the losses of soil carbon (up to 35% in the RCP 4.5 scenario). To avoid any disturbance to the propagation of the seasonal temperatures, short term simulations should use a minimum subsurface thickness of 40 m, even if by the square root law previously described for long temperature trends, a simulation shorter than 10 years would not require more than 30-35 m.

Any LSM, before a simulation starts, must be initialized with appropriate initial conditions, i.e. an initial state of the model

that is close to the real profile at the time. An appropriate initial condition for the temperature of the subsurface is the steady state determined by the surface temperatures at the start of the simulation. This state can be reached during the length of the spinup from arbitrary initial temperatures, if the depth of the bottom boundary is much shallower than the depth determined by the relation $d \propto \sqrt{\kappa t}$, being $t$ the length of the spinup. However, if we increase $d$ enough to prevent the bottom boundary from affecting the thermal diffusion during the length of the simulation, we may also prevent those arbitrary initial temperatures

reaching a steady state during the length of the spinup. This problem can be avoided if the spinup does not start with arbitrary initial subsurface temperatures, but instead from a temperature profile as close as possible to the steady state determined by the surface temperature and linear temperature gradient in Eq. (6).

The temperature differences due to insufficient depth of the bottom boundary and lack of basal heat flux are considerable throughout the subsurface. However, they are of little importance for the global heat budget, as the heat absorbed by the

continents is less than even the uncertainty on the heat absorbed by the oceans (Rhein et al., 2013). The most important

consequences are those on the carbon pools and fluxes in the North Hemisphere. These effects are quantitatively small, and a 1-4% error in the soil carbon pool by 2300 is hardly a first order problem for CLM4.5, compared to the errors introduced by other land systems such as the hydrology model, or the uncertainties on climate projections themselves. However, adding a crustal heat flux and increasing the thickness of the model are computationally cheap and easy to implement. and they will

be necessary to avoid higher errors in the stability of reserves of deep carbon if they are included in the model. In addition to the small errors caused by assuming too shallow lower boundary, we observe large differences when decreasing subsurface thickness to 3.8 m. These results confirm that LSMs should never use subsurface thickness inferior to 30-40 m, as previously concluded by several studies (Alexeev et al., 2007; Nicolsky et al., 2007; Lawrence et al., 2008). Thus, we suggest the use of crustal heat flux and a subsurface thickness enough to avoid these errors, 200 m for a 400 yr simulation, in centennial

simulations of LSMs that include deep carbon deposits. For the correct determination of near-surface permafrost after 400 yr in LSMs, the minimum required subsurface thickness is 40-50 m, but it can be increased to 200 m to avoid errors on the order of 1-4%.

## 6   Conclusions

The area loss of intermediate-depth permafrost (below 42.1 m) in the 1901-2300 period is largely overestimated (3-6 times

larger, for the scenarios RCP 4.5 and RCP 8.5 respectively) for a subsurface of 42.1 m than a subsurface of 342.1 m. Therefore, the inclusion of deep carbon deposits in LSMs will require the use of an appropriate subsurface thickness ($\sim$ 200 m for a 400 yr simulation). The thickness $d$ needed increases with the length $t$ of the simulation following the square-root relation $d \propto \sqrt{\kappa t}$ that was obtained from the theoretical analysis (Stevens et al., 2007), therefore we can rely on theoretical estimates of the optimal depth.

To correctly determine near-surface permafrost in LSMs, the subsurface requires a minimum thickness of 40 m, which avoids bottom boundary effects on the propagation of the seasonal temperature cycle. We suggest the use of a subsurface thickness of 200 m in 400 yr simulations (100 m for 100 yr) and the use of the crustal heat flux, to avoid errors on the order of 1-4%. These changes will be most relevant in centennial simulations of LSMs that include deep carbon deposits.

## Code availability

The modified CLM4.5 software, as well as the instructions for its use in a functional CLM4.5 installation, are available in the Zenodo repository (https://zenodo.org/record/1420497) under the doi 10.5281/zenodo.1420497 (Hermoso de Mendoza, 2018).

## Data availability

The dataset used to produce the initial conditions used in the simulations can be found in the Zenodo repository (https:// zenodo.org/record/1420497) under the doi 10.5281/zenodo.1420497 (Hermoso de Mendoza, 2018). Implementation of these

initial conditions requires modifications to the software, which can be found in the same package.

Three datasets are used as boundary conditions for the simulations (i.e. the atmospheric datasets used to force the land model). The CRUNCEP dataset used to force the model between 1901-2005 is available in the NCAR-UCAR Research Data Archive (Viovy, 2018). The two datasets used to force the model between 2006-2300, RCP 4.5 and RCP 8.5, are available in the Earth System Grid repository (Stern, 2013).

## 5 Acknowledgements

This work was supported by a Discovery grant from the Natural Sciences and Engineering Research Council of Canada (NSERC DG 140576948) and by the Canada Research Chair Program (CRC 230687) to H. Beltrami.

Computational facilities were provided by the Atlantic Computational Excellence Network (ACEnet-Compute Canada) with support from the Canadian Foundation for Innovation. H. Beltrami holds a Canada Research Chair in Climate Dynamics. Ignacio Hermoso was funded by graduate fellowships from a NSERC-CREATE Training Program in Climate Sciences based at St. Francis Xavier University and by additional support from the faculty of sciences at UQAM. Andrew H. MacDougall acknowledges support from the NSERC Discovery Grant program.

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
