# Peer review of "Lower boundary conditions in Land Surface Models. Effects on the permafrost and the carbon pools: a case study with CLM 4.5."

_Geoscientific Model Development, 2018_

## Short Comment (SC1) · 13 Dec 2018

Dear authors,

in my role as Executive editor of GMD, I would like to bring to your attention our Editorial version 1.1:

http://www.geosci-model-dev.net/8/3487/2015/gmd-8-3487-2015.html

This highlights some requirements of papers published in GMD, which is also available on the GMD website in the 'Manuscript Types' section:

http://www.geoscientific-model-development.net/submission/manuscript_types.html

[Figure]

In particular, please note that for your paper, the following requirements have not been met in the Discussions paper:

- "The main paper must give the model name and version number (or other unique identifier) in the title."

- "If the model development relates to a single model then the model name and the version number must be included in the title of the paper. If the main intention of an article is to make a general (i.e. model independent) statement about the usefulness of a new development, but the usefulness is shown with the help of one specific model, the model name and version number must be stated in the title. The title could have a form such as, "Title outlining amazing generic advance: a case study with Model XXX (version Y)"."

Therefore please add a reference to CLM 4.5 in the title of your article in your revised submission to GMD. E.g., "Lower boundary conditions in Land Surface Models. Effects on the permafrost and the carbon pools: a case study with CLM 4.5"

Yours,

Astrid Kerkweg

––––––––––––––––––––––––––––––––

---

## Referee Comment (RC1) · Anonymous Referee #1 · 8 Jan 2019

This paper addresses a very interesting topic. The land surface components of Earth System Models usually make two fundamental simplifications in the model used for computing subsurface temperatures: 1) the geothermal heat flow is not taken into account, 2) the models have an insufficient depth extent to compute the effects of typical climatic thermal perturbations in the subsurface, without being affected by the lower thermal boundary condition. The effects, of both simplifying assumptions are addressed in this paper, focusing specifically on permafrost evolution and the storage/release of carbon in vegetation and soil. The subject of the paper is not new, as the authors acknowledge on page 2, but the effects have thus far hardly been quantified. However, the authors have not provided a full description of their permafrost/thermal

model. Are phase transitions incorporated? Do they couple active layer thickness changes to the hydrology model? What is their definition of permafrost in terms of ice-water content? How are the blanketing and buffering effects of snow on the surface incorporated? Many such descriptions are missing. In addition, the authors assume a constant regolith thickness of a few meters, without porosity-depth changes, and a granitic bedrock to occur worldwide. Also, they assume a spatially constant geothermal heat flow. Both assumptions are very crude approximation of reality, which will severely affect their modelling results. Information on the global variation in subsurface composition and geothermal heat flow is available in literature and databases.

Please find more comments in the supplement

Please also note the supplement to this comment:
https://www.geosci-model-dev-discuss.net/gmd-2018-233/gmd-2018-233-RC1-supplement.pdf

―――――――――――――――――――――

**Supplement:**

Does the paper address relevant scientific modelling questions within the scope of GMD? Yes

Does the paper present a model, advances in modelling science, or a modelling protocol that is suitable for addressing relevant scientific questions within the scope of EGU? Yes

Does the paper present novel concepts, ideas, tools, or data? Partly

Does the paper represent a sufficiently substantial advance in modelling science? Partly

Are the methods and assumptions valid and clearly outlined? Partly

Are the results sufficient to support the interpretations and conclusions? Partly

Is the description sufficiently complete and precise to allow their reproduction by fellow scientists (traceability of results)? In the case of model description papers, it should in theory be possible for an independent scientist to construct a model that, while not necessarily numerically identical, will produce scientifically equivalent results. Model development papers should be similarly reproducible. For MIP and benchmarking papers, it should be possible for the protocol to be precisely reproduced for an independent model. Descriptions of numerical advances should be precisely reproducible. No

Do the authors give proper credit to related work and clearly indicate their own new/original contribution? Yes

Does the title clearly reflect the contents of the paper? The model name and number should be included in papers that deal with only one model. No, see my suggestion

Does the abstract provide a concise and complete summary? Yes

Is the overall presentation well-structured and clear? Yes

Is the language fluent and precise? Yes

Are mathematical formulae, symbols, abbreviations, and units correctly defined and used? Yes

Should any parts of the paper (text, formulae, figures, tables) be clarified, reduced, combined, or eliminated? Yes

Are the number and quality of references appropriate? Yes

Is the amount and quality of supplementary material appropriate? For model description papers, authors are strongly encouraged to submit supplementary material containing the model code and a user manual. For development, technical, and benchmarking papers, the submission of code to perform calculations described in the text is strongly encouraged. NA

This paper addresses a very interesting topic. The land surface components of Earth System Models usually make two fundamental simplifications in the model used for computing subsurface temperatures: 1) the geothermal heat flow is not taken into account, 2) the models have an insufficient depth extent to compute the effects of typical climatic thermal perturbations in the subsurface, without being affected by the lower thermal boundary condition. The effects, of both simplifying assumptions are addressed in this paper, focusing specifically on permafrost evolution and the storage/release of carbon in vegetation and soil. The subject of the paper is not new, as the authors acknowledge on page 2, but the effects have thus far hardly been quantified.

However, the authors have not provided a full description of their permafrost/thermal model. Are phase transitions incorporated? Do they couple active layer thickness changes to the hydrology model? What is their definition of permafrost in terms of ice-water content? How are the blanketing and buffering effects of snow on the surface incorporated? Many such descriptions are missing.

In addition, the authors assume a constant regolith thickness of a few meters, without porosity-depth changes, and a granitic bedrock to occur worldwide. Also, they assume a spatially constant geothermal heat flow. Both assumptions are very crude approximation of reality, which will severely affect their modelling results. Information on the global variation in subsurface composition and geothermal heat flow is available in literature and databases. See Kitover et al. (2014, 2015) for inspiration.

I am not familiar with modeling carbon content changes. Thus I have little comments on those sections.

specific comments:

Title: change to a title better reflecting the contents of the manuscript

e.g.: Effects of geothermal heat flow and assumed model thickness on permafrost distribution and carbon pool changes

page 2:

using the word "reflect" for the thermal effect of a too shallow lower boundary condition can only apply to the effects of climate warming. However, models are also used to study implications of climatic cooling (in the past).

l. 22: 20 C/km is a bit low for a general, global geothermal gradient. 30 C/km is more in line with observations

page 4:

l.8: mention that the two parallel planes are the upper and lower surface

l. 16: assuming a constant diffusivity implies that you assume no porosity change with depth (which is unrealistic for the modeled depth interval), and that no phase change occurs (no melting or freezing). Both assumptions are crude simplifications.

page 7:

l. 4: yes, but porosity decreases exponentially with depth. Thus the thermal diffusivity should change with depth, and is not a constant as you assume.

l. 5: this is a crude assumption. Also composition in the upper 41 meters changes with depth, due to porosity change

l. 7: the assumption that all bedrock (below 41 meters) consists of granite is not realistic

l. 13: mention that you later on will modify the model by incorporating a geothermal heat flow at the base of the model

page 9:

l. 2: you should look better. Such database do exist. For inspiration, check the papers by Kitover et al. (2014, 2015).

l. 6: what is the ice/water content for your permafrost definition? Please note that some authors have advocated a thermal definition of permafrost (like your definition of active layer thickness), since some permafrost in fact lacks ice. Also, please not that some permafrost contains more ice than just the normal porosity (i.e. in the forms of cracks and lenses)

page 29:

l. 35: no, permafrost will also melt from below. The phase transition will affect heat balance and thermal properties of the frozen/unfrozen bedrock. But, the ice-content in bedrock pores and fractures will be low.

page 30:

l. 1-2: yes, but increasing the cell size will reduce the resolution of tracing the lower boundary of the permafrost

technical corrections:

page 1:

l.9.: "… under forcings of two…."

l.13.: use "20 mW/m2" instead of "0.02 W/2"

l.14: replace "frontier" by "interface"

page 2:

l. 29: remove one "the"  (leading to decay of)

page 3:

l.25: replace " is" by "in"

page 5:

l. 2: Insert "Thus"

page 8:

l. 6 please use 50 m instead of 5000 mm

l. 6 what is the relation between the hydrology model (50 meters) and the thermal model (42.1) meters). How are these linked? In the lines above I get the impression that they are coupled for the upper 3.8 meters. But how about the rest?

page 9:

l. 15: of the top of the permafrost

page 15:

l. 9/10: what do you mean? It should affect the amount of heat being diffused

page 22:

l. 11/12: please explain why his happens

page 29:

l. 13: the virtual aquifer has a thickness of 50 meters, not 5

l. 26: ..as high as 50-80% with respect to…

l. 35: no, permafrost will also melt from below. The phase transition will affect heat balance and thermal properties of the frozen/unfrozen bedrock. But, the ice-content in bedrock pores and fractures will be low.

page 30:

l. 1-2: yes, but increasing the cell size will reduce the resolution of tracing the lower boundary of the permafrost

---

## Referee Comment (RC2) · David Lawrence (Referee) · 3 Apr 2019

This paper evaluates the influence of modeling decisions regarding the depth of the soil. It finds that with shallower soils, the influence of the bottom no heat flux boundary can be detected on century timescales.

The study is pretty straightforward and the conclusions are essentially as expected. There are several other papers that have examined a similar topic (Alexeev et al, 2007, Nicolsky et al, 2007, and Lawrence et al., 2008). From my reading of this paper, in comparison to what I recall about these other papers, I think that there is some new information here, but I would strongly recommend that the authors strive to make it

clear how their study is distinct from these previous studies (e.g., global versus site level assessment).

I don't have many technical concerns with the paper. It is fairly straightforward. Run the model at varying soil depths and with and without geothermal heat flux and assess the impact on simulations. The authors covered issues that I would be worried about regarding spinup and computational costs. My main recommendations, in addition to that mentioned above, are:

1. The paper only assesses the impact of extending the depth of ground beyond the default 42m used in CLM4.5. For more context, it would be very useful to also include a simulation with much shallower ground (e.g., 3.5m or so) as is used in most current generation ESMs. My guess, based on the above cited studies, is that the impact of going from 3.5m to 42m is much larger than going from 42m to 342m. That is an important message that needs to be maintained. I wouldn't say that every analysis in the paper needs to be repeated with this shallower version, though for the sake of consistency, it might be worth considering, but for at least the baseline big issues (impact on near-surface permafrost), it should be shown/discussed.

2. There are way too many figures, perhaps even an excess of a factor of 2. Many figures are included that essentially show no change. That doesn't need to be shown in a figure and can easily be characterized in text or a table. The authors should carefully consider each figure and ask whether or not this figure is needed to tell the story. If it isn't required, then remove it, keeping in mind that if the story is that the impact is small (which is part of the story), then that can be stated in words

3. Finally, I think the authors need to carefully consider what their main messages are and, in parallel, put these messages into into context. Currently, they dutifully report about the % change (down to tenths of a percent in many cases) that arises from a deeper column. From my perspective, in the grand scheme of things in Earth System Modeling today, errors of order 1-2% out to 2100 or 2300 are not first order problems.

[Figure]

Uncertainties in climate projections and many other simulated land processes are likely having a much bigger impact on permafrost simulations than the depth of the ground column (once you get beyond a depth of 30m or so). If the authors want to argue otherwise, that's fine, or they can acknowledge that these deep depths may only be relevant on very long timescales or for very specific quantiies. To this end, I would like to see something more in the form of recommendations. An example recommendation could be that if the main interest is in projections of intermediate-depth permafrost thaw, then a deep ground column is required, but if the main interest is in near-surface permafrost, a depth of roughly 50m may be sufficient (and necessary).

Minor points:

1. The reference for CLM4.5 is not Bonan (et al. 2013), it should be Oleson et al. (2013).

2. P.4, line 18: Kirtman et al. is not the correct reference. Kirtman lead the near-term decadal prediction chapter, not the long term projections chapter of AR5.

3. The key reference for the soil biogeochemistry in CLM4.5 is Koven et al. (2013)

4. P.9, line 25: This sentence is not quite correct. Glaciers are represented in CLM4.5 as columns of ice (42m thick, as with the soil). In CESM2, there is the option to run with an ice sheet model beneath CLM, but even in that situation, CLM is still representing the surface mass balance over glaciers and then passing that information to the ice sheet model.

5. One thing that might be worth considering with respect to impact is what the impact might be from having a deep column on the vulnerability of yedoma (not treated in CLM, but with variable soil depths introduced into CLM5, could potentially could be). Yedoma is located deeper in the soil column 5-20m (?) and therefore may be susceptible to the specified soil thickness.

6. Figure 18: You have to study this figure very hard to see the differences. Maybe it
should be removed or difference maps should be shown instead of mean states.

7. P.29, line 12-14. The correct references for variable soil thickness in CLM5 are Brunke et al., 2016 and Swenson and Lawrence (2015)

Nicolsky D. J., V. E. Romanovsky, V. A. Alexeev, D. M. Lawrence, 2007. Improved modeling of permafrost dynamics in a GCM land-surface scheme. Geophys. Res. Lett., 34, L08501, doi.org/10.1029/2007GL029525. Alexeev V. A., D. J. Nicolsky, V. E. Romanovsky, D. M. Lawrence, 2007. An evaluation of deep soil configurations in the CLM3 for improved representation of permafrost, Geophys. Res. Lett., 34, L09502, doi.org/10.1029/2007GL029536. Lawrence, D.M., A.G. Slater, V.E. Romanovsky, and D.J. Nicolsky, 2008. The sensitivity of a model projection of near-surface permafrost degradation to soil column depth and inclusion of soil organic matter. J. Geophys. Res., 113, F02011, doi.org/10.1029/2007JF000883.

Oleson, K.W., D.M. Lawrence, G.B. Bonan, B. Drewniak, M. Huang, C.D. Koven, S. Levis, F. Li, W.J. Riley, Z.M. Subin, S.C. Swenson, P.E. Thornton, A. Bozbiyik, R. Fisher, E. Kluzek, J.-F. Lamarque, P.J. Lawrence, L.R. Leung, W. Lipscomb, S. Muszala, D.M. Ricciuto, W. Sacks, Y. Sun, J. Tang, Z.-L. Yang, 2013. Technical Description of version 4.5 of the Community Land Model (CLM). NCAR Technical Note NCAR/TN-503+STR, doi.org/10.5065/D6RR1W7M.

Koven, C.D., W.J. Riley, Z.M. Subin, J.-Y. Tang, M.S. Torn, W.D. Collins, G.B. Bonan, D.M. Lawrence, and S.C. Swenson, 2013. The effect of vertically-resolved soil biogeo-chemistry and alternate soil C and N models on C dynamics of CLM4. Biogeosciences, 10, doi.org/10.5194/bg-10-7109-2013.

Swenson, S.C. and D.M. Lawrence, 2015. GRACE-based assessment of interannual variability in groundwater simulated in the Community Land Model. Water Res. Res., 51, 8817-8833, doi.org/10.1002/2015WR017582.

Brunke, M.A., P. Broxton, J. Pelletier, D. Gochis, P. Hazenberg, D.M. Lawrence, L.R.

Leung, G.-Y. Niu, P.A. Troch, and X. Zeng, 2016. Implementing and evaluating variable soil thickness in the Community Land Model, version 4.5 (CLM4.5). J. Climate, 29, doi.org/10.1175/JCLI-D-15-0307.1.

---

## Author Response (AR1)

Dear authors,

in my role as Executive editor of GMD, I would like to bring to your attention our Editorial version 1.1:

http://www.geosci-model-dev.net/8/3487/2015/gmd-8-3487-2015.html

This highlights some requirements of papers published in GMD, which is also available on the GMD website in the 'Manuscript Types' section:

http://www.geoscientific-model-development.net/submission/manuscript_types.html

[Figure]

In particular, please note that for your paper, the following requirements have not been met in the Discussions paper:

- "The main paper must give the model name and version number (or other unique identifier) in the title."

- "If the model development relates to a single model then the model name and the version number must be included in the title of the paper. If the main intention of an article is to make a general (i.e. model independent) statement about the usefulness of a new development, but the usefulness is shown with the help of one specific model, the model name and version number must be stated in the title. The title could have a form such as, "Title outlining amazing generic advance: a case study with Model XXX (version Y)"."

Therefore please add a reference to CLM 4.5 in the title of your article in your revised submission to GMD. E.g., "Lower boundary conditions in Land Surface Models. Effects on the permafrost and the carbon pools: a case study with CLM 4.5"

Yours,

Astrid Kerkweg
* * *
[Figure]

Geosci. Model Dev. Discuss.,
https://doi.org/10.5194/gmd-2018-233-RC1, 2019

[Figure]

This paper addresses a very interesting topic. The land surface components of Earth System Models usually make two fundamental simplifications in the model used for computing subsurface temperatures: 1) the geothermal heat flow is not taken into account, 2) the models have an insufficient depth extent to compute the effects of typical climatic thermal perturbations in the subsurface, without being affected by the lower thermal boundary condition. The effects, of both simplifying assumptions are addressed in this paper, focusing specifically on permafrost evolution and the storage/release of carbon in vegetation and soil. The subject of the paper is not new, as the authors acknowledge on page 2, but the effects have thus far hardly been quantified. However, the authors have not provided a full description of their permafrost/thermal

model. Are phase transitions incorporated? Do they couple active layer thickness changes to the hydrology model? What is their definition of permafrost in terms of ice-water content? How are the blanketing and buffering effects of snow on the surface incorporated? Many such descriptions are missing. In addition, the authors assume a constant regolith thickness of a few meters, without porosity-depth changes, and a granitic bedrock to occur worldwide. Also, they assume a spatially constant geothermal heat flow. Both assumptions are very crude approximation of reality, which will severely affect their modelling results. Information on the global variation in subsurface composition and geothermal heat flow is available in literature and databases.

Please find more comments in the supplement

Please also note the supplement to this comment:
https://www.geosci-model-dev-discuss.net/gmd-2018-233/gmd-2018-233-RC1-supplement.pdf

—————————————————

[Figure]

Does the paper address relevant scientific modelling questions within the scope of GMD? Yes

Does the paper present a model, advances in modelling science, or a modelling protocol that is suitable for addressing relevant scientific questions within the scope of EGU? Yes

Does the paper present novel concepts, ideas, tools, or data? Partly

Does the paper represent a sufficiently substantial advance in modelling science? Partly

Are the methods and assumptions valid and clearly outlined? Partly

Are the results sufficient to support the interpretations and conclusions? Partly

Is the description sufficiently complete and precise to allow their reproduction by fellow scientists (traceability of results)? In the case of model description papers, it should in theory be possible for an independent scientist to construct a model that, while not necessarily numerically identical, will produce scientifically equivalent results. Model development papers should be similarly reproducible. For MIP and benchmarking papers, it should be possible for the protocol to be precisely reproduced for an independent model. Descriptions of numerical advances should be precisely reproducible. No

Do the authors give proper credit to related work and clearly indicate their own new/original contribution? Yes

Does the title clearly reflect the contents of the paper? The model name and number should be included in papers that deal with only one model. No, see my suggestion

Does the abstract provide a concise and complete summary? Yes

Is the overall presentation well-structured and clear? Yes

Is the language fluent and precise? Yes

Are mathematical formulae, symbols, abbreviations, and units correctly defined and used? Yes

Should any parts of the paper (text, formulae, figures, tables) be clarified, reduced, combined, or eliminated? Yes

Are the number and quality of references appropriate? Yes

Is the amount and quality of supplementary material appropriate? For model description papers, authors are strongly encouraged to submit supplementary material containing the model code and a user manual. For development, technical, and benchmarking papers, the submission of code to perform calculations described in the text is strongly encouraged. NA

This paper addresses a very interesting topic. The land surface components of Earth System Models usually make two fundamental simplifications in the model used for computing subsurface temperatures: 1) the geothermal heat flow is not taken into account, 2) the models have an insufficient depth extent to compute the effects of typical climatic thermal perturbations in the subsurface, without being affected by the lower thermal boundary condition. The effects, of both simplifying assumptions are addressed in this paper, focusing specifically on permafrost evolution and the storage/release of carbon in vegetation and soil.  The subject of the paper is not new, as the authors acknowledge on page 2, but the effects have thus far hardly been quantified.

However, the authors have not provided a full description of their permafrost/thermal model. Are phase transitions incorporated? Do they couple active layer thickness changes to the hydrology model? What is their definition of permafrost in terms of ice-water content? How are the blanketing and buffering effects of snow on the surface incorporated? Many such descriptions are missing.

In addition, the authors assume a constant regolith thickness of a few meters, without porosity-depth changes, and a granitic bedrock to occur worldwide. Also, they assume a spatially constant geothermal heat flow. Both assumptions are very crude approximation of reality, which will severely affect their modelling results.  Information on the global variation in subsurface composition and geothermal heat flow is available in literature and databases. See Kitover et al. (2014, 2015) for inspiration.

I am not familiar with modeling carbon content changes. Thus I have little comments on those sections.

specific comments:

Title: change to a title better reflecting the contents of the manuscript

e.g.: Effects of geothermal heat flow and assumed model thickness on permafrost distribution and carbon pool changes

page 2:

using the word "reflect" for the thermal effect of a too shallow lower boundary condition can only apply to the effects of climate warming. However, models are also used to study implications of climatic cooling (in the past).

l. 22: 20 C/km is a bit low for a general, global geothermal gradient. 30 C/km is more in line with observations

page 4:

l.8: mention that the two parallel planes are the upper and lower surface

l. 16: assuming a constant diffusivity implies that you assume no porosity change with depth (which is unrealistic for the modeled depth interval), and that no phase change occurs (no melting or freezing). Both assumptions are crude simplifications.

page 7:

l. 4: yes, but porosity decreases exponentially with depth. Thus the thermal diffusivity should change with depth, and is not a constant as you assume.

l. 5: this is a crude assumption. Also composition in the upper 41 meters changes with depth, due to porosity change

l. 7: the assumption that all bedrock (below 41 meters) consists of granite is not realistic

l. 13: mention that you later on will modify the model by incorporating a geothermal heat flow at the base of the model

page 9:

l. 2: you should look better. Such database do exist. For inspiration, check the papers by Kitover et al. (2014, 2015).

l. 6: what is the ice/water content for your permafrost definition? Please note that some authors have advocated a thermal definition of permafrost (like your definition of active layer thickness), since some permafrost in fact lacks ice. Also, please not that some permafrost contains more ice than just the normal porosity (i.e. in the forms of cracks and lenses)

page 29:

l. 35: no, permafrost will also melt from below. The phase transition will affect heat balance and thermal properties of the frozen/unfrozen bedrock. But, the ice-content in bedrock pores and fractures will be low.

page 30:

l. 1-2: yes, but increasing the cell size will reduce the resolution of tracing the lower boundary of the permafrost

technical corrections:

page 1:

l.9.: "… under forcings of two…."

l.13.: use "20 mW/m2" instead of "0.02 W/2"

l.14: replace "frontier" by "interface"

page 2:

l. 29: remove one "the"  (leading to decay of)

page 3:

l.25: replace " is" by "in"

page 5:

l. 2: Insert "Thus"

page 8:

l. 6 please use 50 m instead of 5000 mm

l. 6 what is the relation between the hydrology model (50 meters) and the thermal model (42.1) meters). How are these linked? In the lines above I get the impression that they are coupled for the upper 3.8 meters. But how about the rest?

page 9:

l. 15: of the top of the permafrost

page 15:

l. 9/10: what do you mean? It should affect the amount of heat being diffused

page 22:

l. 11/12: please explain why his happens

page 29:

l. 13: the virtual aquifer has a thickness of 50 meters, not 5

l. 26: ..as high as 50-80% with respect to…

l. 35: no, permafrost will also melt from below. The phase transition will affect heat balance and thermal properties of the frozen/unfrozen bedrock. But, the ice-content in bedrock pores and fractures will be low.

page 30:

l. 1-2: yes, but increasing the cell size will reduce the resolution of tracing the lower boundary of the permafrost

Geosci. Model Dev. Discuss.,
https://doi.org/10.5194/gmd-2018-233-RC2, 2019

[Figure]

This paper evaluates the influence of modeling decisions regarding the depth of the soil. It finds that with shallower soils, the influence of the bottom no heat flux boundary can be detected on century timescales.

The study is pretty straightforward and the conclusions are essentially as expected. There are several other papers that have examined a similar topic (Alexeev et al, 2007, Nicolsky et al, 2007, and Lawrence et al., 2008). From my reading of this paper, in comparison to what I recall about these other papers, I think that there is some new information here, but I would strongly recommend that the authors strive to make it

clear how their study is distinct from these previous studies (e.g., global versus site level assessment).

I don't have many technical concerns with the paper. It is fairly straightforward. Run the model at varying soil depths and with and without geothermal heat flux and assess the impact on simulations. The authors covered issues that I would be worried about regarding spinup and computational costs. My main recommendations, in addition to that mentioned above, are:

1. The paper only assesses the impact of extending the depth of ground beyond the default 42m used in CLM4.5. For more context, it would be very useful to also include a simulation with much shallower ground (e.g., 3.5m or so) as is used in most current generation ESMs. My guess, based on the above cited studies, is that the impact of going from 3.5m to 42m is much larger than going from 42m to 342m. That is an important message that needs to be maintained. I wouldn't say that every analysis in the paper needs to be repeated with this shallower version, though for the sake of consistency, it might be worth considering, but for at least the baseline big issues (impact on near-surface permafrost), it should be shown/discussed.

2. There are way too many figures, perhaps even an excess of a factor of 2. Many figures are included that essentially show no change. That doesn't need to be shown in a figure and can easily be characterized in text or a table. The authors should carefully consider each figure and ask whether or not this figure is needed to tell the story. If it isn't required, then remove it, keeping in mind that if the story is that the impact is small (which is part of the story), then that can be stated in words

3. Finally, I think the authors need to carefully consider what their main messages are and, in parallel, put these messages into into context. Currently, they dutifully report about the % change (down to tenths of a percent in many cases) that arises from a deeper column. From my perspective, in the grand scheme of things in Earth System Modeling today, errors of order 1-2% out to 2100 or 2300 are not first order problems.
[Figure]

Uncertainties in climate projections and many other simulated land processes are likely having a much bigger impact on permafrost simulations than the depth of the ground column (once you get beyond a depth of 30m or so). If the authors want to argue otherwise, that's fine, or they can acknowledge that these deep depths may only be relevant on very long timescales or for very specific quantiies. To this end, I would like to see something more in the form of recommendations. An example recommendation could be that if the main interest is in projections of intermediate-depth permafrost thaw, then a deep ground column is required, but if the main interest is in near-surface permafrost, a depth of roughly 50m may be sufficient (and necessary).

Minor points:

1. The reference for CLM4.5 is not Bonan (et al. 2013), it should be Oleson et al. (2013).

2. P.4, line 18: Kirtman et al. is not the correct reference. Kirtman lead the near-term decadal prediction chapter, not the long term projections chapter of AR5.

3. The key reference for the soil biogeochemistry in CLM4.5 is Koven et al. (2013)

4. P.9, line 25: This sentence is not quite correct. Glaciers are represented in CLM4.5 as columns of ice (42m thick, as with the soil). In CESM2, there is the option to run with an ice sheet model beneath CLM, but even in that situation, CLM is still representing the surface mass balance over glaciers and then passing that information to the ice sheet model.

5. One thing that might be worth considering with respect to impact is what the impact might be from having a deep column on the vulnerability of yedoma (not treated in CLM, but with variable soil depths introduced into CLM5, could potentially could be). Yedoma is located deeper in the soil column 5-20m (?) and therefore may be susceptible to the specified soil thickness.

6. Figure 18: You have to study this figure very hard to see the differences. Maybe it

should be removed or difference maps should be shown instead of mean states.

7. P.29, line 12-14. The correct references for variable soil thickness in CLM5 are Brunke et al., 2016 and Swenson and Lawrence (2015)

Nicolsky D. J., V. E. Romanovsky, V. A. Alexeev, D. M. Lawrence, 2007. Improved modeling of permafrost dynamics in a GCM land-surface scheme. Geophys. Res. Lett., 34, L08501, doi.org/10.1029/2007GL029525. Alexeev V. A., D. J. Nicolsky, V. E. Romanovsky, D. M. Lawrence, 2007. An evaluation of deep soil configurations in the CLM3 for improved representation of permafrost, Geophys. Res. Lett., 34, L09502, doi.org/10.1029/2007GL029536. Lawrence, D.M., A.G. Slater, V.E. Romanovsky, and D.J. Nicolsky, 2008. The sensitivity of a model projection of near-surface permafrost degradation to soil column depth and inclusion of soil organic matter. J. Geophys. Res., 113, F02011, doi.org/10.1029/2007JF000883.

Oleson, K.W., D.M. Lawrence, G.B. Bonan, B. Drewniak, M. Huang, C.D. Koven, S. Levis, F. Li, W.J. Riley, Z.M. Subin, S.C. Swenson, P.E. Thornton, A. Bozbiyik, R. Fisher, E. Kluzek, J.-F. Lamarque, P.J. Lawrence, L.R. Leung, W. Lipscomb, S. Muszala, D.M. Ricciuto, W. Sacks, Y. Sun, J. Tang, Z.-L. Yang, 2013. Technical Description of version 4.5 of the Community Land Model (CLM). NCAR Technical Note NCAR/TN-503+STR, doi.org/10.5065/D6RR1W7M.

Koven, C.D., W.J. Riley, Z.M. Subin, J.-Y. Tang, M.S. Torn, W.D. Collins, G.B. Bonan, D.M. Lawrence, and S.C. Swenson, 2013. The effect of vertically-resolved soil biogeo-chemistry and alternate soil C and N models on C dynamics of CLM4. Biogeosciences, 10, doi.org/10.5194/bg-10-7109-2013.

Swenson, S.C. and D.M. Lawrence, 2015. GRACE-based assessment of interannual variability in groundwater simulated in the Community Land Model. Water Res. Res., 51, 8817-8833, doi.org/10.1002/2015WR017582.

Brunke, M.A., P. Broxton, J. Pelletier, D. Gochis, P. Hazenberg, D.M. Lawrence, L.R.

[Figure]

Leung, G.-Y. Niu, P.A. Troch, and X. Zeng, 2016. Implementing and evaluating variable soil thickness in the Community Land Model, version 4.5 (CLM4.5). J. Climate, 29, doi.org/10.1175/JCLI-D-15-0307.1.

———————————————

[Figure]

The Executive editor has brought to our attention that the title of the main paper does not fulfill the following requirements for papers published in GMD:

- "The main paper must give the model name and version number (or other unique identifier) in the title."
- "If the model development relates to a single model then the model name and the version number must be included in the title of the paper. If the main intention of an article is to make a general (i.e. model independent) statement about the usefulness of a new development, but the usefulness is shown with the help of one specific model, the model name and version number must be stated in the title. The title could have a form such as, "Title outlining amazing generic advance: a case study with Model XXX (version Y)"."

Therefore, we will change the title of the paper to include a reference to CLM4.5. We have decided to take the title suggested by the Executive editor: "Lower boundary conditions in Land Surface Models. Effects on the permafrost and the carbon pools: a case study with CLM 4.5".

Best regards.

We thank the reviewer for his comments, which show that several points both in the description of the model and in the objectives and limitations of our study needed to be clarified. We have made some corrections and added several paragraphs to address the reviewer's questions. In addition, we have made many editorial corrections throughout the manuscript to improve the readability and flow of the text. We have also changed the Figure 18 (P21) to show the differences to the original model and changed its color code to make it colorblind-friendly. In response to a suggestion made by reviewer 2, we have also moved several figures to supplementary materials. We now provide a response to all the comments and concerns expressed by the reviewer.

1. **However, the authors have not provided a full description of their permafrost/thermal model. Are phase transitions incorporated? Do they couple active layer thickness changes to the hydrology model? What is their definition of permafrost in terms of ice-water content? How are the blanketing and buffering effects of snow on the surface incorporated? Many such descriptions are missing.**

    As requested by the reviewer, we have added qualitative descriptions for the snow model and the hydrology model within the Community Land Model version 4.5 (CLM4.5) in the subsection 3.1 "Original Land Model". The hydrology model parameterizes interception, throughfall, canopy drip, snow accumulation and melt, water transfer between snow layers, infiltration, evaporation, surface runoff, subsurface drainage, redistribution within the soil column, and groundwater discharge and recharge. The vertical movement of water in the soil is determined by hydrological properties of the soil layers, which can be altered by their ice content as increased ice content reduces the effective porosity of the soil. The model also implements an artificial aquifer with a capacity of 5000 mm at the bottom of the soil column, from which discharge is calculated. The parameterization of snow consists of up to 5 layers, whose number and thickness increase with the thickness of the snowpile. Thermal conduction in these layers works like in soil layers, with the thermal properties of ice and water. The model includes fractional snow cover and phase transitions between the ice and water in the soil and snow layers. We have not included the full numerical description of the snow and hydrology models, because they can be found in the technical description paper for CLM4.5. The only explicit numerical description is that for the layer scheme in CLM4.5 and the zero heat flux condition used at the bottom boundary, because these are the only parts of the numerical model that we modify.

    In the subsection 3.4 "Permafrost treatment", we have added the commonly used definition of permafrost as the ground that remains below 0C for two consecutive years. This is a thermal definition of permafrost, i.e. the permafrost is defined only by the temperature of a layer, without regard that the layer actually contains ice. This allows our definition to also apply in bedrock layers, where the numerical model does not include water. Permafrost in the soil, to which we refer in the paper as near-surface permafrost, hinders the infiltration of liquid water from upper layers because the ice fills all pores, reducing the effective porosity of a permafrost layer to zero.

2. **In addition, the authors assume a constant regolith thickness of a few meters, without porosity-depth changes, and a granitic bedrock to occur worldwide. Also, they assume a spatially constant geothermal heat flow. Both assumptions are very crude approximation of reality, which will severely affect their modelling results. Information on the global variation in subsurface composition and geothermal heat flow is available in literature and databases.**

    The assumptions of constant regolith thickness and global granitic bedrock were not made by us, but by the modeling group who developed CLM4.5. We pointed in the paper that the homogeneity of the subsurface and other characteristics of the subsurface model in CLM4.5 are very unrealistic assumptions which affect the thermal state of the subsurface and the hydrology model. However, the goal of this paper is not to make precise predictions with a detailed model of the subsurface including soil composition and thickness, bedrock properties and heat flow variations because the data to build such a model do not exist. Our aim is to investigate and quantify the effects of two unrealistic assumptions made by most land models, i.e. the zero value for the geothermal heat flux and the

excessive thinness of the model's subsurface, and to this end we modified CLM4.5. Including fine variations in the composition of the bedrock or thickness of the soil is maybe desirable, but is simply not possible at the spatial resolution of the model because the data are too sparse, and it is outside the scope of this paper.

We agree that using a spatially constant geothermal heat flow is a very crude approximation of reality. However, it allows us to treat the basal heat flow as a parameter which we can increase at regular intervals between 0 (the basal heat flux value used in CLM4.5) and 80 mW/m2, in order to quantify the effect of basal heat flow in CLM4.5 within a range of values of heat flow in stable continents. Likewise, we have systematically changed the thickness of the modeled subsurface in order to demonstrate how the use of a too shallow model affects the energy budget of the subsurface. Maps of geothermal heat flow are available in literature, however these maps are in large part extrapolated from an incomplete data set with many regions void of data, in particular in permafrost regions where these data are most important (Jaupart and Mareschal, 2015). Kitover et al. (2014, 2015) used a map made by Davies and Davies (2010), who extrapolated the data on the basis of crude correlation between geology and heat flux, which leaves a large uncertainty on the mean heat flux for each cell. Wide regions of the globe remain void of measurements of geothermal heat flow, in particular the high-latitude regions.

3. **Please also note the supplement to this comment: https://www.geosci-model-dev-discuss.net/gmd-2018-233/gmd-2018-233-RC1- supplement.pdf**

The supplement to the reviewer's comment states that our description is not sufficiently complete and precise to allow its reproduction. We respectfully disagree. The Community Earth System Model version 1.2 (CESM1.2), which includes the CLM4.5, is released to the public and can be easily found in the website of the University Corporation for Atmospheric Research (UCAR). The paper states explicitly what changes we have made to the numerical model. To reproduce our simulations, one only needs to modify the CLM4.5 codes to program the same changes as ours and run the simulations using the same forcing data. These modifications are described in the paper and the specific code changes are available in the Zenodo repository, as specified in the section "Code availability". The initial state of the model for the simulations is provided in the same Zenodo repository, and we have described the spinup process that it is used to drive the CLM4.5 to this state from arbitrary initial conditions. Finally, the forcing data are publicly available, with references provided in the section "Data availability". Therefore, the paper provides all the information necessary to allow the reproduction of our results.

As stated in the supplement, the title does not include the model name and number. This has already been pointed out in a previous comment, and will be corrected in the final version of the paper. The new name will be "Lower boundary conditions in Land Surface Models. Effects on the permafrost and the carbon pools: a case study with CLM4.5".

We now address point by point the list of specific comments of the reviewer:
- **Using the word "reflect" for the thermal effect of a too shallow lower boundary condition can only apply to the effects of climate warming. However, models are also used to study implications of climate cooling (in the past).** In the mathematical formulation for the propagation of a surface signal (a wave) into the subsurface, the lower boundary acts by bouncing the signal (with strength damped across the slab of subsurface bounded between the surface and the lower boundary) back to the surface, effectively "reflecting" the signal. This applies to any signal regardless of its sign, therefore we do not understand why would the word "reflect" not be valid for cooling signals, while being appropriate only for warming signals
- **20 C/km is a bit low for a general, global geothermal gradient. 30 C/km is more in line with observations.** We beg to disagree on this point. Mean continental heat flux is 60 mW/m2 and conductivity of bedrock in the model is 3 W/m/K, which gives a geothermal gradient of 20 K/km. Among all the gradients measured in the Canadian Shield, most are between 10 and 15K/km, and none is higher than 15K/km (Jaupart et al., 2015). Similar observations have been reported over all Precambrian and Paleozoic provinces worldwide.
- **Mention that the two parallel planes are the upper and lower surface.** We have made this correction.

- **Assuming a constant diffusivity implies that you assume no porosity change with depth (which is unrealistic for the modeled depth interval), and that no phase change occurs (no melting or freezing). Both assumptions are crude simplifications.** We agree that this is a crude simplification. However, this is a theoretical calculation where we want to show what the difference that a subsurface of 342.1m as opposed to the 42.1m would make in CLM4.5. In CLM4.5, only the upper 3.8m of the subsurface models hydrology and implements some degree of heterogeneity in its thermal or hydraulic properties. In this simplified calculation, we consider it is acceptable to model the upper 3.8 m as having the same homogeneous granitic composition as the subsurface below, as our goal with this rough calculation is to provide justification to the experiments we perform afterwards with several CLM4.5 versions of increased subsurface thickness.
- **Porosity decreases exponentially with depth. Thus the thermal diffusivity should change with depth, and is not a constant as you assume. Composition in the upper 31 meters changes with depth, due to porosity change. The assumption that all bedrock (below 41 m) consists of granite is not realistic.** In the subsection 3.1 "Original Land Model", we limit ourselves to describe the composition, properties and layout of the subsurface scheme in CLM4.5. While we agree that these assumptions in CLM4.5 are very crude approximations of reality, the objective of this paper is not to correct them.
- **Mention that you later on will modify the model by incorporating a geothermal heat flow at the base of the model.** We have added this mention.
- **Such database (of geothermal heat flow) do exist. For inspiration, check the papers by Kitover et al. (2014, 2015).** We are aware of the existence of the heat flow map used in Kitover et al. (2014, 2015). This map was produced by Davies and Davies (2010) and is based on the same heat flow database as that used by Jaupart and Mareschal (2015), using a different methodology and interpolation method. The heat flow measurements, as we stated in the paper, do not cover wide areas of Canada, Siberia, the Middle East, Africa and South America. To create the global map, Davies and Davies (2010) used a correlation between geology and geothermal heat flux to extrapolate in these void areas, which leads to very poor estimates in the areas with no measurements.
- **What is the ice/water content for your permafrost definition? Please note that some authors have advocated a thermal definition of permafrost since some permafrost in fact lacks ice. Also, please note that some permafrost contains more ice than just the normal porosity (i.e. in the forms of cracks and lenses).** We have now added the definition of permafrost in the subsection 3.4 "Permafrost treatment", and defines permafrost as the ground that remains below 0C for two consecutive years, which is indeed a thermal definition of permafrost. In addition to near-surface permafrost (defined for the depth range where the soil extends), we also define intermediate-depth permafrost to cover the portion of the subsurface composed of impermeable bedrock, therefore we believe taht a thermal definition is appropriate. Also, while we are aware that permafrost ice can be contained in interstitial spaces such as cracks and lenses, these are regrettably not defined in the subsurface model for CLM4.5.
- (... the only process taking place in bedrock is thermal diffusion.) **No, permafrost will also melt from below. The phase transition will affect heat balance and thermal properties of the frozen/unfrozen bedrock. But, the ice content in bedrock pores and fractures will be low.** While in reality bedrock holds water, in CLM4.5 (and most land models) bedrock is modeled as not having any water content at all. As such, bedrock layers in CLM4.5 only include thermal diffusion processes, both in and out of the permafrost region. For this reason, we stated that adding more of such bedrock layers to the land model would carry very small computational costs.
- (... if we keep the original scheme where layer thickness increase exponentially, it is possible to increase the thickness of the model to hundreds of meters by adding only a few layers.) **Yes, but increasing the cell size will reduce the resolution of tracing the lower boundary of the permafrost.** We agree, the exponential layer thickness scheme decreases the resolution of the permafrost depth range. This already shows in CLM4.5, as the bottom soil layer has a thickness of 1.5 m, out of a total soil thickness of 3.8 m. However, we think that the exponential scheme used in CLM4.5 is appropriate, because it allows to increase the depth of the model easily. While resolution is important, it is necessary to find a tradeoff between resolution and computational cost, which was the original reason behind the design of the exponential layer thickness by the Community modeling group. This balance between resolution and simplicity can be expressed though the scaling factor for the exponential node depth formula described in Eq. (7), so this parameter could be adjusted to meet a better

compromise between resolution and computational performance. We have added a mention of this concern in the discussion.

In addition to his comments, the reviewer has also made a series of technical corrections for whose we are very grateful. We have corrected the typos and made the text corrections in the reviewer's list. We have addressed the other corrections (with the exception of the two last points, which are repeated in the previous list of specific comments) in the following list:

- **Use "20mW/m2" instead of "0.02W/m2".** As requested, we have changed all the units from Watts to mili-Watts throughout the text.

- **Please use 50 m instead of 5000 mm.** We assume the reviewer means 5 m instead of 50 m. The technical description paper of CLM4.5 used "mm" as the units for water capacity (per unit area), including the explicit use of "5000 mm" as the capacity of this aquifer, which is why we kept these units. As this is not a matter of big importance, we have changed "5000 mm" to "5 m" throughout the paper.

- **What is the relation between the hydrology model (50 m) and the thermal model (42.1 m) How are these linked? In the lines above I get the impression that they are coupled for the upper 3.8 m. But how about the rest?** As we stated in the paper, the aquifer (with a capacity of 5 m, not 50 m) exists as a virtual layer below the soil. It is not coupled for the upper 3.8 m, it is a layer below this depth. To clarify what we call "virtual", we added the explanation in the text: it is a layer that does not interact with the subsurface other that to store water. This is, while it should physically occupy the same space as the bedrock in the subsurface model, it simply takes all the water that percolates from the bottom soil layer without this water affecting the thermal properties of the bedrock or being affected by phase transitions, and then it send the water directly to the river transport model. As we pointed out in the discussion, this model is completely unrealistic, but fortunately ithis has been addressed in the new CLM5.0 version.

- **What do you mean? It should affect the amount of heat being diffused.** By "the magnitude of the heat flux used as bottom boundary condition does not affect heat diffusion" we mean that thermal diffusivity is independent of temperature. Therefore, in a purely conductive regime, the heat equation is linear and the temperature anomaly solution for the propagation of a thermal signal into the subsurface can be superposed to the steady state solution (determined by the non-anomaly initial temperature and the geothermal gradient) This implies that heat diffusion (the transient part of the solution) is not affected by the value of the steady state heat flux. This can be verified in Carslaw and Jaeger (1959) "Conduction of heat in solids".

- (... Increasing the crustal heat flux decreases the initial concentration of soil carbon in some areas while increasing it in others.) **Please explain why this happens.** The local variability of the results across the Northern Hemisphere permafrost region is difficult to interpret with certainty, so we have added an plausible explanation in the discussion, rather than in the results section. The possible explanation is that the increasing the subsurface temperature decreases the period of seasonal freezing for some soil layers, which allows more methane to be produced if there is still a frozen soil layer beneath, which restricts the seepage of water and allowis the active layer to be inundated. However if the entirety of the soil thaws, the water can percolate to the aquifer and less methane is produced. Because the differences in the methane production accumulate over time, this also explains the local differences in the size of the carbon pool. Similarly, the presence of more liquid water allows for a slightly larger vegetation growth while the percolation of water to the aquifer decreases it. The maps for soil carbon, vegetation carbon and methane production match with what we should expect from this explanation: the first situation happens in coldest areas, where the lowermost soil layers remain frozen, and the second situation occurs in the periphery of the permafrost region, where the lowest layer can thaw.

- **The virtual aquifer has a thickness of 50 m, not 5.** As we explained before, this is incorrect. The virtual aquifer has a capacity for 5 m of water.

Jaupart C., Labrosse S., Lucazeau F., Mareschal J.-C. (2015). Temperatures, Heat and Energy in the Mantle of the Earth, in *Treatise on Geophysics, 2nd Edition, vol. 7, The Mantle*, edited by D, Bercovici, 223-270, Elsevier.

We thank the reviewer for his comments, which show that we need to better put the article into context and emphasize its main conclusions. We have made many editorial corrections, including the bibliographic mistakes, and added several paragraphs to address the reviewer's questions.

1. **There are several other papers that have examined a similar topic (Alexeev et al, 2007, Nicolsky et al, 2007, and Lawrence et al., 2008). From my reading of this paper, in comparison to what I recall about these other papers, I think that there is some new information here, but I would strongly recommend that the authors strive to make it clear how their study is distinct from these previous studies (e.g., global versus site level assessment).**

   We have added a paragraph in the introduction, to explain the differences between our study and those mentioned by Dr. Lawrence. To improve the modeling of permafrost, the papers mentioned by Dr. Lawrence pointed out that the subsurface model must be thick enough (at least 30 m) to capture the damping of the annual surface temperature. These papers increased the thickness of the CLM3 from 3.5 m to different depths to capture decadal and centennial variability during the 20[th] century. Alexeev et al. (2007) used of a slab of variable thickness (30, 100 and 300 m) at the bottom of a several layers representing the soil with high resolution, in order to have sufficient depth to absorb decadal to centennial signals. Nicolsky et al. (2007) did the same by using additional soil layers to increase the thickness of the model to 80 m, which they applied at specific locations with deep permafrost. Lawrence et al. (2008) tried depths up to 125 m by adding extra bedrock layers, and determined how this affected the extent of near-surface permafrost. These studies did not consider crustal heat flux and although they studied the impacts of model depth in near-surface permafrost, they did not analyze the associated effects to the permafrost carbon pool. In our paper, we look into the impacts of the thickness of the subsurface and the crustal heat flux, not only on permafrost but also on the heat content of the subsurface and on the carbon pool, in simulations for the 20[th] century that we continue until 2300 under two scenarios of anthropogenic emissions.

2. **The paper only assesses the impact of extending the depth of ground beyond the default 42m used in CLM4.5. For more context, it would be very useful to also include a simulation with much shallower ground (e.g., 3.5m or so) as is used in most current generation ESMs. My guess, based on the above cited studies, is that the impact of going from 3.5m to 42m is much larger than going from 42m to 342m. That is an important message that needs to be maintained. I wouldn't say that every analysis in the paper needs to be repeated with this shallower version, though for the sake of consistency, it might be worth considering, but for at least the baseline big issues (impact on near-surface permafrost), it should be shown/discussed.**

   As suggested by Dr. Lawrence, we have included a new simulation with shallow ground (3.8m) by removing the bedrock in the model. We already observed that increasing the thickness of the model provides diminishing returns, therefore, reducing the thickness of the subsurface from 3.5m to 42m has a bigger impact than going from 42m to 342m. The impact of progressively increasing depth depends on the timescale of the simulation, so the increase from 42m to 342m is more significant for a millennial-scale simulation than it is for our centennial-scale simulations. In this new simulation, we observe that decreasing the subsurface thickness from 42m to 3.8m has a much larger effect in the soil carbon pool than from increasing it from 42m to 342m. The loss of soil carbon during the 1901-2300 period is increased by 4.4% in the RCP85 scenario, but more importantly by 35% in the RCP4.5 scenario. The emissions of methane are consistently 1-2% higher for a subsurface of 3.8m than one of 42m, which results in these increased losses of soil carbon. It has already been well established that deepening the bottom boundary below 3.5m improves representation of permafrost significantly, bringing the simulated extent of present permafrost much closer to the observations (Alexeev et al, 2007, Nicolsky et al, 2007, and Lawrence et al., 2008, Koven et al., 2013, Slater & Lawrence, 2013). We have not detected a significant decrease in the areal extent of near-surface permafrost, but decreasing model thickness from 42m to 3.8m affects the thickness and depth of permafrost. We

have included this point in the discussion, and we have also emphasized the logical conclusion that can be inferred from the diminishing returns to subsurface thickness and the optimal depths. Increasing subsurface thickness produces modest improvements, but reducing it introduces serious miscalculations to subsurface temperature, permafrost and soil carbon.

3. **There are way too many figures, perhaps even an excess of a factor of 2. Many figures are included that essentially show no change. That doesn't need to be shown in a figure and can easily be characterized in text or a table. The authors should carefully consider each figure and ask whether or not this figure is needed to tell the story. If it isn't required, then remove it, keeping in mind that if the story is that the impact is small (which is part of the story), then that can be stated in words.**

We agree that the number of figures is too large, and we have reduced it significantly. Following the recommendation of the reviewer, we have removed from the main paper many figures that show very small changes and that can be sufficiently explained in the text or with the support of the tables. These figures have been moved to supplementary materials, which we will submit along with the revised version of the paper. We have moved Figures 9, 10, 11, 12, 14, 15, 17, 23, 24, 28 and 29, cutting the number of figures in the main body of the paper from 29 to 18. We have kept Figure 16 although it shows only a small difference, to have at least one figure showing the evolution of near-surface permafrost, and Figures 19 and 20, because they show the significant differences produced by the crustal heat flux to the evolution of the soil carbon pool. We have also changed Figure 18 significantly, to show the differences to the original model in the same way as Figures 21, 25 and 27 do. We have also eliminated the 2000 CE time frame in Figures 18, 21, 25 and 27, which allows us to enlarge these maps.

4. **Finally, I think the authors need to carefully consider what their main messages are and, in parallel, put these messages into into context. Currently, they dutifully report about the % change (down to tenths of a percent in many cases) that arises from a deeper column. From my perspective, in the grand scheme of things in Earth System Modeling today, errors of order 1-2% out to 2100 or 2300 are not first order problems. Uncertainties in climate projections and many other simulated land processes are likely having a much bigger impact on permafrost simulations than the depth of the ground column (once you get beyond a depth of 30m or so). If the authors want to argue otherwise, that's fine, or they can acknowledge that these deep depths may only be relevant on very long timescales or for very specific quantities. To this end, I would like to see something more in the form of recommendations.**

We agree that the order of these errors are small compared to other sources of error, and we will not argue otherwise. We however defend that these small errors are very easily avoidable, because the implementation of a crustal heat flux and the extension of subsurface thickness is justified, easy to implement and computationally cheap. We have added a new paragraph at the end of the discussion, where we acknowledge the small scale of the corrected errors, but at the same time arguing our point. We also acknowledge that it is more important to not drop subsurface thickness below 40m than to extend it to 200m, but that the importance of a thick subsurface increases with the time scale of the simulation. We provide a explicit recommendation to have a subsurface thickness of at least 40-50m for a correct reproduction of near-surface permafrost, and increase it to 200 m to avoid errors in the order of 1-4%, even more if we were to include deep carbon deposits in the model.

The reviewer also made several minor points, which we address point by point:
1. **The reference for CLM4.5 is not Bonan (et al. 2013), it should be Oleson et al. (2013).** We have corrected this reference.
2. P.**4, line 18: Kirtman et al. is not the correct reference. Kirtman lead the near-term decadal prediction chapter, not the long term projections chapter of AR5.** We have corrected this reference with Collins et al., 2013 (Climate Change 2013: The physical Science Basis. Long-term Climate Change: Projections, Commitments and irreversibility).
3. **The key reference for the soil biogeochemistry in CLM4.5 is Koven et al. (2013).** We have corrected this reference.
4. **P.9, line 25: This sentence is not quite correct. Glaciers are represented in CLM4.5 as columns of ice (42m thick, as with the soil). In CESM2, there is the option to run with an ice sheet model beneath CLM, but even in that situation, CLM is still representing the surface mass balance over glaciers and then passing that information to the ice**

**sheet model.** We have corrected this sentence. It now states that CLM4.5 represents the interior of Greenland with the upper 42 m of ice and passes this information to the land-ice model, but it does not represent the soil.

5. **One thing that might be worth considering with respect to impact is what the impact might be from having a deep column on the vulnerability of yedoma (not treated in CLM, but with variable soil depths introduced into CLM5, could potentially be). Yedoma is located deeper in the soil column 5-20m (?) and therefore may be susceptible to the specified soil thickness.** We have added Yedoma and frozen thermokarst deposits as an example of deep carbon deposits in the discussion. These hold an estimated 211 +/- 160 PgC of carbon in depths up to 50 m (Strauss et al., 2013). Our study shows that the thawing of intermediate-depth permafrost is largely overestimated by the 42 m subsurface, therefore an appropriate subsurface thickness of 200m would be necessary if these deep carbon deposits were included in the model.

6. **Figure 18: You have to study this figure very hard to see the differences. Maybe it should be removed or difference maps should be shown instead of mean states.** We have changed this figure to show the active layer thickness of the original CLM4.5 and the differences between the modified versions of the model and the original model. We have also changed the color scale to be colorblind-friendly.

7. **P.29, line 12-14. The correct references for variable soil thickness in CLM5 are Brunke et al., 2016 and Swenson and Lawrence (2015).** We have corrected these references.

Strauss, J., Schirrmeister, L., Grosse, G., Wetterich, S., Ulrich, M., Herzschuh, U., & Hubberten, H. W. (2013). The deep permafrost carbon pool of the Yedoma region in Siberia and Alaska. *Geophysical Research Letters*, *40*(23), 6165-6170.

Koven, C. D., Riley, W. J., & Stern, A. (2013). Analysis of permafrost thermal dynamics and response to climate change in the CMIP5 Earth System Models. *Journal of Climate*, *26*(6), 1877-1900.

Slater, A. G., & Lawrence, D. M. (2013). Diagnosing present and future permafrost from climate models. *Journal of Climate*, *26*(15), 5608-5623.

**Author's changes to the manuscript "Lower boundary conditions in Land Surface Models. Effects on the permafrost and the carbon pools."**

Dear Editor,

To address the comments from the referees, we have made numerous corrections to the manuscript. We provide a list of the changes done to the manuscript, following the order of the referees' comments. To facilitate the task of the Topical Editor, we have pointed each modification to its specific location in the marked-up version of the manuscript, that highlights the changes made. In addition, we have corrected typographical mistakes and made many minor corrections throughout the manuscript to improve the readability of the text. Because these corrections are too numerous, we do not include them in a list to avoid making this cover letter tedious, but they can be easily seen throughout the marked-up version of the manuscript that has been produced with latexdiff for LaTeX.

Before discussing the list of specific changes, we would like to bring to the attention of the Editor the most important modifications to the content and the structure of the manuscript:

- In response to the comment of the Executive Editor of GMD, we have changed the title of the manuscript to that he suggested for us: "Lower boundary conditions in Land Surface Models. Effects on the permafrost and the carbon pools: a case study with CLM4.5."

- To answer the concern from Reviewer #2 that the manuscript had too many figures, we have moved 11 figures to a supplementary materials file: Figures 9, 10, 11, 12, 14, 15, 17, 23, 24, 28 and 29 from the original manuscript, which have been renamed Figures S1 to S11 in the supplementary materials. Figures 13, 16, 18, 19, 20, 21, 22, 25, 26 and 27 have been relabeled Figures 9 to 18 in the new version of the manuscript.

- As suggested by Reviewer #2, we have added a new simulation (made with a modified version of CLM4.5 that uses a subsurface 3.8 m thick) to those presented in the original manuscript. Consequently, we refer to this simulation and its results in the introduction, results and discussion sections. The results of this simulation have also been added to the Tables 1, 2 and 4, and to the Figures 6, 7, 10, 11, 12, 14, 16, 18, S1, S8 and S10.

- We have split Section 5 "Discussion and Conclusions" into two separate sections: Section 5 "Discussion" and Section 6 "Conclusions".

The Reviewer #1 (anonymous) pointed out in his general comments that several explanations were lacking in the manuscript. He also provided a list of specific comments, which included the points made in his general comments. We made several corrections and additions to the manuscript to address these concerns:

1. Section 2 "Theoretical analysis", P4, lines 24-25. We have clarified that the 2 parallel planes are the surface and the lower boundary.

2. Subsection 3.1 "Original land model", P8, lines 1-2. We have corrected the statement that the thermal properties of the soil are also affected by the soil water content, not only by carbon density.

3. Subsection 3.1 "Original land model", P8, line 2. We have clarified that the bedrock in CLM4.5

does not allow for pores or interstices where water can be held.

4. Subsection 3.1 "Original land model", P8, lines 9-10. We have added a reminder that we will afterwards modify the model to include geothermal heat flux.

5. Subsection 3.1 "Original land model", P8, lines 11-22. We have added two new paragraphs with the qualitative descriptions of the snow and hydrology models used in CLM4.5.

6. Subsection 3.4 "Permafrost treatment", P10, lines 17-19. We have added a paragraph to provide an explicit definition of permafrost.

7. Section 5 "Discussion", P30, lines 7-13. We have added a paragraph where we defend our reasons to use a uniform heat flux, and we also explain why we did not modify some of the most simplistic assumptions of the model such as global granitic bedrock and constant regolith depth. In this paragraph we also argue that the existing maps of heat flux, bedrock composition and soil thickness are incomplete.

8. Section 5 "Discussion", P31, lines 21-24. We have added two sentences exposing the concerns of the reviewer of how the exponential layer scheme decreases the resolution of permafrost depth, and our thoughts on the usefulness of the exponential scheme despite these drawbacks.

The Reviewer #1 also provided another list of technical corrections. To address these comments, including the text corrections, we have made the following changes:

1. P1, line 9. In the sentence, "under two future scenarios" has been corrected to "under forcings of two future scenarios".

2. The units used for heat flux have been changed from $W/m^2$ to $mW/m^2$. We have applied this correction throughout the manuscript, as well as all Figures and Tables.

3. P1, line 15. We have replaced "soil-bedrock frontier" by "soil-bedrock interface". We have also corrected a mistake in the increased temperature: from 0.4 K to 0.04 K.

4. P2, line 34. In "leading to the decay of" we have removed "the".

5. P4, line 12. We have corrected "the general solution is the time derivative" to "the general solution in the time derivative".

6. P5, line 13. We have inserted "Thus" at the start of the sentence.

7. P9, line 14. The capacity of the unconfined aquifer in CLM4.5 has been changed from 5000mm to 5m.

8. We have added a clarification of how the aquifer works in the new paragraph describing the hydrological model at the end of the subsection 3.1 "Original land model", P8, lines 14-17. We have also added a small note to remind this in the subsection 3.2 "Carbon model", P9, lines 14-15.

9. P10, line 30. In the sentence, "the maximum depth of permafrost" has been corrected to "the maximum depth of the top of the permafrost".

10. Section 5 "Discussion", P30, line 34 to P31, line 9.  We have added a paragraph where we provide an explanation for the regional variability observed in the results for methane production, soil carbon and vegetation carbon.

11. P28, line 5. In the sentence, we have changed "can be within 50-80% of that of" to "can be as high as 50-80% with respect to".

The Reviewer #2 (Dr. David Lawrence) made very useful comments showing that our work needed to be differentiated from previous studies, and made several recommendations such as better putting the work into context, reducing the number of figures in the manuscript, and exploring the effects of reducing the subsurface thickness in the model with a new simulation. In response to his comments, we have made the following changes:

1.  Section 1 "Introduction", P3, lines 17-24. We have added a few sentences to refer to 3 papers that explored the increase of the subsurface thickness in CLM3, and to explain the innovation in our paper relative to these previous studies.

2.  We have included a new simulation with a modified model of subsurface thickness 3.8m. Consequently, we have added the results of this simulation to the figures, tables, and the text in Section 4 "Results". We also added associated mentions to this simulation in the introduction and the description of our changes to the models, and we added a discussion of the new results in Section 5 "Discussion".

3.  Section 5 "Discussion", P31, lines 25-35. We have added a new paragraph where we discuss the diminishing returns of increasing the thickness of the subsurface, and how the impact of going from 42m to 3.8m is far more important that going from 42m to 342m. We relate this result with the previous studies with CLM3, where subsurface thickness was increased from 3.5m to more than 30m to improve the simulation of permafrost.

4.  We have moved 11 figures from the Section 4 "Results" to a supplementary materials file (Figures 9, 10, 11, 12, 14, 15, 17, 23, 24, 28 and 29 from the original text). The references to these figures throughout the text have been changed to the remaining figures and tables, which are enough to support the exposed results.

5.  Figures 18, 21, 25 and 27 (now renumbered to Figures 11, 14, 16 and 18). We have eliminated the second column corresponding to the 2000 CE frame, and we added a new row corresponding to the new modified model with 3.8m subsurface thickness. Because now the figures are taller than they are wide, we have changed the orientation of these figures back to portrait (previously it had been changed to landscape because of these figures were wider than they were tall).

6.  Section 5 "Discussion", P32, lines 12-27. We added a paragraph were we acknowledge the small scale of the errors derived from using a subsurface of 42m while we argue for the convenience of increasing the thickness of the model nonetheless. Based on our results, including the large errors observed for a subsurface of 3.8m, we also provide recommendations for the subsurface thickness that LSMs should use, in relation to the time scale of the simulations. These recommendations have also been added to the new Section 6 "Conclusions" in P33, lines 4-8.

The Reviewer #2 also provided a list of minor points, which are very relevant. To address them, we have made the following modifications to the manuscript:

1.  We have corrected the reference Oleson et al. (2013), which was mistakenly Bonan et al. (2013) throughout the text (the leading authors Oleson & Lawrence were missing in the author list).

2.  Section 2 "Theoretical Analysis" P5, line 9. We have replaced Kirtman et al. (2013), which makes reference to the short-term predictions in IPCC 2013, by the correct reference for long term projections, Collins et al. (2013),

3. Subsection 3.4 "Carbon model", P8, line 25. We have added the reference Koven et al. (2013) for the BioGeoChemical Cycles in Biome-BGC.

4. Subsection 3.4 "Permafrost treatment", P11, lines 8-9. We have corrected the statement where we wrongly said that Greenland is not included in CLM4.5. Now it correctly states that even though CLM4.5 does not represent the soil below the Greenland ice sheet, it represents the upper 42m of the ice sheet.

5. Section 5 "Discussion", P30, lines 18-23. We have included yedoma and frozen thermokarst deposits as an example of deep carbon deposits (up to 50m deep), and discussed the implications that the inclusion of these deposits would have in the land model, on view of our results for intermediate-depth permafrost.

6. Figure 18 in the original manuscript (Figure 11 in the new version) has been changed to show the differences relative to the original CLM4.5 model for the modified versions. We have also changed the color code in this figure to be colorblind-friendly.

7. Section 5 "Discussion", P30, lines 1-6. We have added the references Swenson and Lawrence (2015) and Brunke et al. (2016) for variable soil thickness in CLM5. We have also added the references Pelletier et al. (2016) and Clair et al. (2015) for measurements and global estimations of soil thickness.

We thank again the referees for their comments, which have allowed us to seriously improve our paper. The marked-up version of the manuscript follows,

Ignacio Hermoso de Mendoza

[revised manuscript text omitted]

---

## Author Response (AR2)

**Author's changes to the manuscript "Lower boundary conditions in Land Surface Models. Effects on the permafrost and the carbon pools: a case study with CLM 4.5."**

Dear Editor,

To address the comments of the referees, we have made several changes in the manuscript, mainly in the Abstract, Introduction, Discussion and Conclusion sections. To facilitate the task of the Topical Editor, we provide a marked-up version of the manuscript produced with latexdiff for LaTeX, which highlights the changes between the revised manuscript and the previous version seen by the reviewers. Changes in Figures are not marked by latexdiff, but these are limited to the order of the legends in Figures 6, 10 and 12 of the manuscript and Figures S1, S8 and S10 in the Supplementary Materials.

Reviewer #1 made a few corrections and minor comments, that we would like to address here:

1. **The authors had one more simulation with 3.8 m thickness, but when I compared these results, the difference from 3.8 m to 42 m was larger than other simulations. Although the author claims that this difference depends on timescale, but I think this also has important implication for short time simulations. You may want to point it out in the abstract as well as in the discussion part.**

   We agree with the Reviewer that the implication for short term simulations is important and should be pointed out in the abstract. Therefore, we have indicated in the abstract that reducing thickness from 42.1 m to 3.8 m has a larger effect than increasing it to 342.1 m, because a subsurface as thin as 3.8 m affects the propagation of the seasonal temperature cycle. For this reason we recommend short term simulations to use a subsurface of at least 40 m. We discussed the larger effect of reducing thickness to 3.8 m and its reasons, but we now include the explicit recommendation to keep minimum thickness to 40 m even in short-term simulations. We have added this in the Discussion and the Conclusion, as well as in the summary of our findings in the Abstract.

2. **The author missed the 3.8 m simulation results in the Figure 8 and Figure 9?**

   Figures 8 and 9 show respectively the heat stored within the upper 42.1 m and the area where permafrost is present within the upper 42.1 m. Therefore, it made no sense to us to include the 3.8 m simulation in these figures, as the portion of the subsurface shown in these figures does not exist for this simulation.

3. **Also maybe rearrange the legend in Figure 6 and 10. Right now it starts from 42 to 342 and then 3.8.**

   We have rearranged the legend and the caption of the Figures 6 and 10, to be ordered by increasing d. We made the same change to Figure 12, as well as Figures S1, S8 and S10 in the Supplementary Materials.

4. **Methane is mainly controlled by the soil temperature and inundation, I am more interested in how these variables compare with the CLM 4.5 default settings, since results shows the heat fluxes are quite different among these simulations**

   It is difficult to show meaningful figures indicating the differences in soil temperature and

inundation between our different versions of CLM4.5. In the case of soil temperature, it is easy to show temperature-depth profiles for a token cell of the land model, at single moment in time. To show results for the whole land system, or to show the dependency in time of the temperature differences that arise between simulations, it is more useful to show the amount of heat contained in the subsurface, which can be easily added for different portions of the subsurface and for several cells. The Figures S1, S2 and S4 from the Supplementary Materials, which show the differences of heat content for the soil, are the best figures that display a global picture of the thermal differences between the simulations, for the portion of the subsurface where methane is produced. For inundation, it is challenging to extrapolate a direct measurement of the exact time/portion of the soil that is inundated during our simulations because there is no direct output from CLM4.5 for this. In the discussion we theorize that the frozen soil layers may create inundations above the water table, so that knowledge of the water table is not sufficient. The best we can do is to compare soil moisture content in the active layer. However, we only have knowledge of this on the monthly basis used for the output of our simulations, without information of how active layer thickness or soil moisture vary in the meantime.

Reviewer #2 also made a few comments. We have addressed them in the following paragraphs:

1. **Thanks to the authors response, it is now much clearer to me what the goal of this study is, as it is now clearly stated in response #2: "[the] aim is to investigate and quantify the effects of two unrealistic assumptions made by most land models". Maybe the authors can use this clear formulation in the Introduction?**

   We agree with the reviewer in that we should be as clear as possible in our goals. We have added this formulation in the last paragraph of the Introduction, where we state the objectives of our study.

2. **The authors make quite a few and important assumptions in their modeling, like a low and constant geothermal gradient everywhere on the globe (also in areas not consisting of Precambrian and Paleozoic 'provinces'), a granitic crustal composition everywhere, constant regolith/soil thickness, regolith/soil with constant porosity, etc., which likely affect the results to a large extent. The authors added a paragraph in the Discussion (page 30) where the modeling simplifications and the reasons for making them are mentioned. But that is much too late in the manuscript; it should be at the start of the Discussion, and it should also be mentioned in the Abstract and in the Introduction.**

   It is important to remark that the mentioned assumptions (constant regolith thickness, uniform bedrock composition and properties, etc) are not our own assumptions, but that of the CLM4.5. Because these are neither our assumptions nor the object of our study, their place is not the Abstract or the Introduction but the description of the CLM4.5 in the Methodology. For the same reason, they are only discussed for how they could affect the results obtained for the two model parameters that are the focus of this paper (the crustal heat flux and the thickness of the subsurface) and therefore they are not the focus of the Discussion. Our only assumption is the uniformity of the geothermal gradient, which is mentioned in the Abstract and the Introduction, and justified in the Methodology section and in the Discussion. We consider that assumptions intrinsic to the CLM4.5 should not be presented in the Abstract or the Introduction as our own, because that would lead to confusion about the extent of our modifications to this model.

3. **Furthermore, I would appreciate it if the authors would discuss the implications of their simplifications for the simulation predictions.The current statement (by the authors) is**

**that quantification of the effects is too much effort for them (because it would require recoding and more simulations). Furthermore, the authors state that maps for heat flow etc are incomplete. But incomplete maps are better than no maps. If the authors do not want/need to spend time on better research, they can try to discuss the impacts of their simplifications qualitatively.**

The implications that the simplifications assumed by the CLM4.5 may have in our results are discussed qualitatively in paragraphs 7, 8 and 9 of the Discussion. If the reviewer is referring to the uniformity of the heat flux, which is an assumption made by the authors, this is discussed in paragraph 8. Incomplete maps are only useful in the regions where they have information, but they are no better than no maps in the regions devoid of data. Unfortunately, there are almost no heat flow measurements in the permafrost region that is the object of our study because a temperature probe cannot be lowered in a borehole filled with ice. A string of thermistors can be left in the hole after drilling and measurements made after the hole has returned to equilibrium but this is very costly. Other solutions such as filling a casing in the hole with kerosene are now prohibited for environmental reasons. Using an extrapolated heat flux map without control points only gives us a result dependent on the assumptions. Instead, varying uniformly the value of the bottom heat flow allows us to establish a quantitative relationship between its value and the effects it has on permafrost and biogeochemistry, and to bracket the range of responses of the subsurface to the climate forcing. We have modified the 8$^{th}$ paragraph of the discussion to make our reasons more clear. We also discuss that while continental heat flow is not uniform, long wavelength variations across stable regions are not very large. Moreover, most permafrost regions in the North Hemisphere belong to PreCambrian shields where heat flow is low and stable. Therefore, we can expect that the results with an extrapolated heat flow map would not be better than those with the uniform heat flow that we used.

We thank again the reviewers for their comments, which have allowed us to improve the clarity of our paper. The marked-up version of the manuscript follows,

Ignacio Hermoso de Mendoza

[revised manuscript text omitted]